# Nanostructured polymer films with metal-like thermal conductivity

Yanfei Xu [1,3], Daniel Kraemer[1,4], Bai Song[1,5], Zhang Jiang[2], Jiawei Zhou [1], James Loomis[1], Jianjian Wang[1,6], Mingda Li[1,7], Hadi Ghasemi[1,8], Xiaopeng Huang[1,9], Xiaobo Li[1,10] & Gang Chen [1]

Due to their unique properties, polymers – typically thermal insulators – can open up opportunities for advanced thermal management when they are transformed into thermal conductors. Recent studies have shown polymers can achieve high thermal conductivity, but the transport mechanisms have yet to be elucidated. Here we report polyethylene films with a high thermal conductivity of 62 Wm$^{-1}$ K$^{-1}$, over two orders-of-magnitude greater than that of typical polymers (~0.1 Wm$^{-1}$ K$^{-1}$) and exceeding that of many metals and ceramics. Structural studies and thermal modeling reveal that the film consists of nanofibers with crystalline and amorphous regions, and the amorphous region has a remarkably high thermal conductivity, over ~16 Wm$^{-1}$ K$^{-1}$. This work lays the foundation for rational design and synthesis of thermally conductive polymers for thermal management, particularly when flexible, lightweight, chemically inert, and electrically insulating thermal conductors are required.

[1] Department of Mechanical Engineering, Massachusetts Institute of Technology, Cambridge, MA 02139, USA. [2] Advanced Photon Source, Argonne National Laboratory, Argonne, IL 60439, USA. [3] Present address: Department of Mechanical and Industrial Engineering, University of Massachusetts Amherst, Amherst, MA 01003, USA. [4] Present address: Modern Electron, Bothell, WA 98011, USA. [5] Present address: Department of Energy and Resources Engineering, Peking University, Beijing 100871, China. [6] Present address: Advanced Cooling Technologies, Inc., Lancaster, PA 17601, USA. [7] Present address: Department of Nuclear Science and Engineering, Massachusetts Institute of Technology, Cambridge, MA 02139, USA. [8] Present address: Department of Mechanical Engineering, University of Houston, Houston, TX 77004, USA. [9] Present address: 2205 W Olive Way, Chandler, AZ 85248, USA. [10] Present address: State Key Laboratory of Coal Combustion, School of Energy and Power Engineering, Huazhong University of Science and Technology, 430074 Wuhan, Hubei, China. Correspondence and requests for materials should be addressed to G.C. (email: gchen2@mit.edu)

From soft robotics, organic electronics to 3D printing and artificial skin, polymers continue to infiltrate modern technologies thanks to their unique combination of properties not available from any other known materials[1–5]. They are lightweight, durable, flexible, corrosion resistant, and easy to process, and hence are expected to offer significant advantages over traditional heat conductors, such as metals and ceramics[1]. However, application of polymers in thermal management has been largely hampered by their low thermal conductivities (~0.1 $Wm^{-1} K^{-1}$)[6]. To date, metals and ceramics remain the dominant heat conductors.

The fact that polyethylene (0.2–0.5 $Wm^{-1} K^{-1}$)[6,7] is composed of a backbone of carbon-carbon bonds similar to those in diamond, one of the most thermally conductive materials (above 1000 $Wm^{-1} K^{-1}$)[8], encourages research in thermally conductive polymers. Importantly, atomistic simulations have suggested that an individual crystalline polyethylene chain can achieve very high —possibly divergent—thermal conductivity[9], in agreement with the non-ergodic characteristics of one-dimensional conductors discussed by Fermi et al.[10]. However, the experimental measurement of such theoretically high thermal conductivities remains elusive. By increasing the crystallite orientation and crystallinity, the thermal conductivity of polymers can increase considerably[11–19], such as polyethylene nanofibers (~104 $Wm^{-1} K^{-1}$)[14]. Although exceptionally conductive, these measured values are still much lower than the numerical predictions for bulk single-crystalline polyethelyene (~237 $Wm^{-1} K^{-1}$)[20,21]. There is no precise mechanism that accounts for the deviation of experimental and theoretical values. And the main factors that govern the thermal conductivity in these fibers remain poorly understood[19]. It is generally known that such materials are not perfect crystals, but instead semicrystalline polymers containing mixed crystalline and amorphous regions[6]. Translating the remarkably high thermal conductivity seen in simulation as well as in polyethylene nanofibers into a scalable polymer presents a major challenge in synthesis. Overcoming this challenge will broadly expand the scope of nanofiber use in thermal management, since practical applications require large areas or volumes of materials[22]. Recently, Ronca et al.[16] reported stretched ultra-high molecular weight film with thermal conductivity as high as 65 $Wm^{-1} K^{-1}$, measured using a commercial laser-flash system, and Zhu et al. reported thermal conductivity of fibers as high as 51 $Wm^{-1} K^{-1}$ by further processing of commercial spectra fibers using an electrothermal method[17]. These reports show the potential of achieving high thermal conductivity in macroscopic samples. However, the structural property relationship has yet to be further elucidated.

We have been engaged in scaling up the high thermal conductivity of individual nanofiber to more macroscale films[23]. Here, we report a thermal conductivity measurement of 62 $Wm^{-1} K^{-1}$ in polyethylene films (Fig. 1). The thermal conductivity in our film outperforms that of many conventional metals (304-stainless steel ~15 $Wm^{-1} K^{-1}$)[24] and ceramics (aluminum oxide ~30 $Wm^{-1} K^{-1}$)[25]. Motivated by the theoretically large thermal conductivity of single-crystal polymer[9,20], we fabricate thermally conductive polymer films with an emphasis on minimally entangling and maximally aligning the chain, rather than solely pursuing a high crystallinity. We further uncover the thermal transport mechanisms through the combination of structural analysis, determined by high-resolution synchrotron X-ray scattering, and a phenomenological thermal transport model. We find that the film actually consists of nanofibers with crystalline and amorphous regions along the fiber and that the amorphous regions have remarkably high conductivity (~16 $Wm^{-1} K^{-1}$), which is central to the high thermal conductivity (~62 $Wm^{-1} K^{-1}$). Increased control over amorphous morphology is a promising route toward achieving thermal conductivities approaching theoretical limits.

## Results

**Polymer processing.** We start with commercial semi-crystalline polyethylene powders (Fig. 1a), which feature randomly oriented lamellar crystallites (lamellae) dispersed in an amorphous chain network (Fig. 1d). We dissolve the powder above its melting temperature in decalin, allowing the initially entangled chains to disentangle (Fig. 1d). This greatly reduces the entanglements for the subsequent processing. Afterward, the hot solution is extruded through a custom-built Couette-flow system[23], which imparts a shear force on the polymer chains and led to further disentanglement[26]. To maintain the disentangled structure, the extruded solution flows directly onto a liquid nitrogen-cooled substrate. Some segments of the polyethylene chains fold back into thin lamellae upon drying[7], while others remain disordered albeit less entangled (Fig. 1d)[27]. Finally, the as-extruded films (Fig. 1b) are mechanically pressed and drawn inside a heated enclosure using a continuous and scalable roll-to-roll system[23]. Heating allows the disentangled polymer chains to move more freely and facilitates alignment along the draw direction (Fig. 1c, d)[6].

**Microscale and nanoscale morphology of polymers.** In order to track the evolution of polymer structures, we imaged the as-purchased powders, the extruded films and films of various draw ratios (final length/initial length) using scanning electron microscopy (SEM, Fig. 1e–j). The powder consists of porous particles with an average size of ~100 μm (Fig. 1e). After extrusion, the film surface appeared isotropic with randomly distributed microflakes (Fig. 1f). During drawing, the film self-organized into a clear fibrous texture along the draw direction. The diameters of the fibers comprising the film reduced as the draw ratio increased, which led to a smoother and denser texture (see ×10 and ×110 in Fig. 1g, h). We further tore a ×70 film apart to explore the detailed internal structures where individual fiber can be clearly observed (Fig. 1i, j), and multiple interior fibers with smaller diameter ~8 nm were also seen (Supplementary Fig. 3).

**Thermal conductivity measurements.** To study the thermal properties of these polyethylene films, we employed two distinct experimental schemes: a home-built steady-state system[28] (Fig. 2a) and a widely-adopted transient method called time domain thermoreflectance (Fig. 2c, Supplementary Notes 2 and 3)[29–31]. On the steady-state platform, we measured heat current as a function of temperature difference across a sample film (Fig. 2b, Supplementary Fig. 1a and Table 1). We investigated the systematic errors including on radiation and parasitic heat losses, and carefully minimized measurement errors (see Supplementary Note 2, Supplementary Figs. 1 and 2). To validate the steady-state measurement accuracy, we tested several control samples including 304-stainless steel foil (S. Steel 304)[24], Zylon[15] fibers, Dyneema[15] fibers, Sn films[25], and Al films[25], our measured thermal conductivity values are in general agreement with established values[15,24,25] (Fig. 2a, Supplementary Fig. 1 and Table 2). We next measured the thermal conductivities of a series of films with various draw ratios (Figs. 2b and 3a and Supplementary Table 1). The as-extruded (×1) film was found to have an in-plane thermal conductivity of 0.38 $Wm^{-1} K^{-1}$. As draw ratio was increased, film thermal conductivity along the draw direction was improved dramatically, reaching 62 $Wm^{-1} K^{-1}$ at ×110 (Figs. 2b and 3a). Notably, we saw no sign of saturation in thermal conductivity (Fig. 3a, b will be discussed below), which suggested more room for further conductivity enhancement beyond

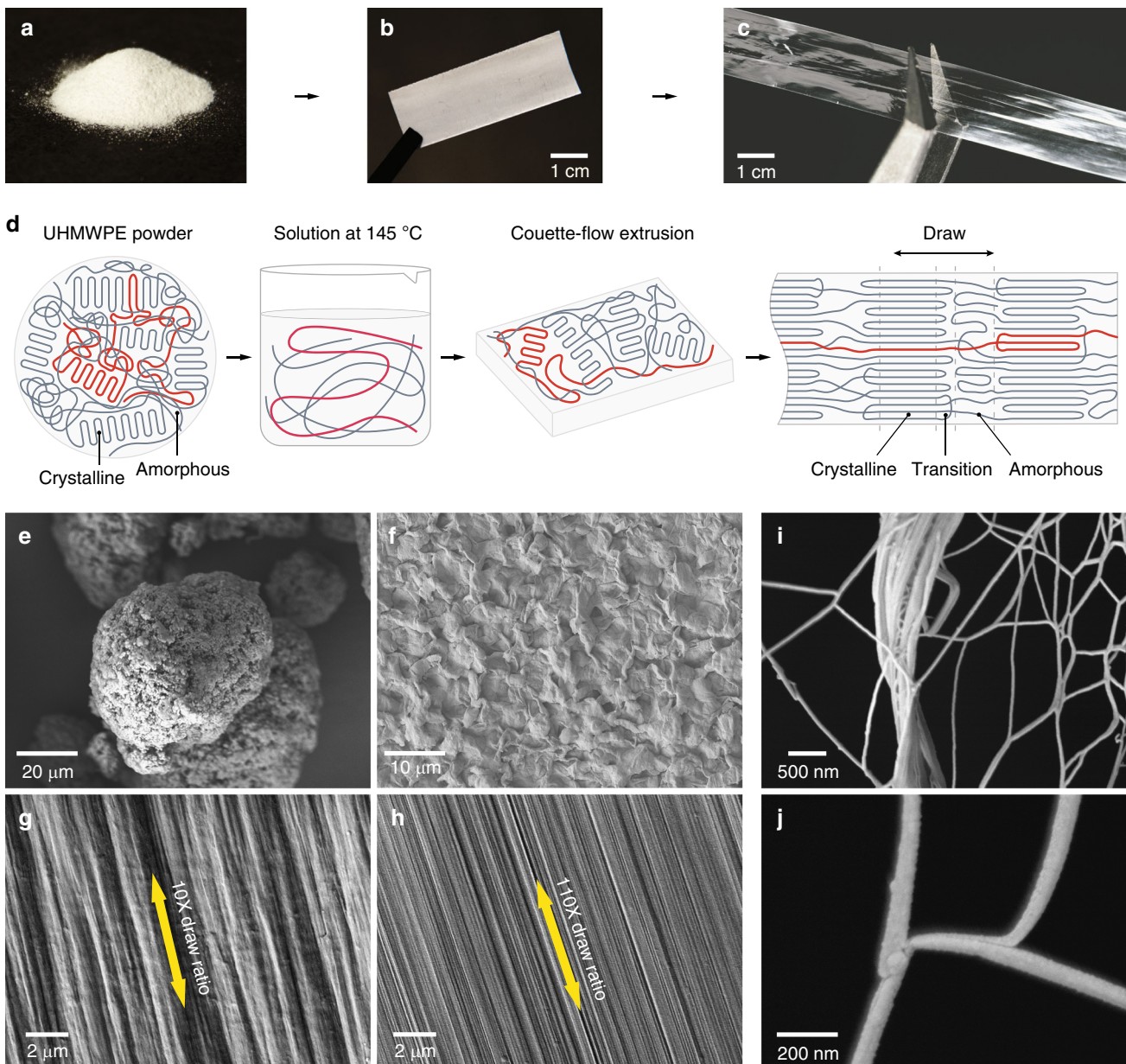

**Fig. 1** Fabrication and characterization of polymer films with high thermal conductivity. **a–c** Photos of commercial ultrahigh molecular weight polyethylene (UHMWPE) powders, a thick opaque as-extruded film and a thin transparent drawn film, respectively. **d** Illustration of film morphology evolution during fabrication. The powders feature lamellar polyethylene crystallites embedded in a disordered and entangled chain network. The degree of entanglement greatly reduces in the hot decalin solution and after subsequent Couette-flow extrusion. The ultradrawn films are characterized by oriented crystallites interconnected by aligned amorphous chains. **e** Scanning electron microscope (SEM) images of some UHMWPE powders. Scale bar indicates 20 μm. **f** SEM image of an as-extruded film. Scale bar indicates 10 μm. **g** SEM image of a 10× draw ratio film. Scale bar indicates 2 μm. **h** SEM image of a ×110 film. Scale bar indicates 2 μm. **i, j** SEM images of a torn ×70 film revealing the polyethylene nanofibers as the basic building blocks. Scale bar indicates 500 nm and 200 nm, respectively

×110 draw ratio. Recent atomistic simulations further corroborate this expectation[20].

Two-color time-domain thermoreflectance (TDTR) experiments were conducted to study transient heat conduction in the films[29–31] and to further validate the steady-state results (Fig. 2b). We fabricated a 150-μm-thick laminate consisting of 100 layers of ×50 films and carefully microtomed a cross-section (roughness ~10 nm, Supplementary Fig. 6 and Note 2) perpendicular to the draw direction. Representative thermoreflectance signals are reported in Fig. 2d, from which we extracted an average thermal conductivity of 33.6 W m⁻¹ K⁻¹ (Fig. 3a) along the draw direction. The TDTR results agree well with values obtained

using the steady-state system (Fig. 3a). The successful demonstration of 100-layer laminate with such high thermal conductivity implies potential scalability not only along the drawing direction, but also in the thickness direction. In addition, we have investigated film thermal stability, obtaining <5% thermal conductivity variations before and after annealing (24 h at 80 °C).

**Atomic scale and nanoscale structure investigation.** To reveal correlation between such a high thermal conductivity and structure, we quantitatively investigated the structure at both atomic scale and nanoscale by high-resolution wide-angle and

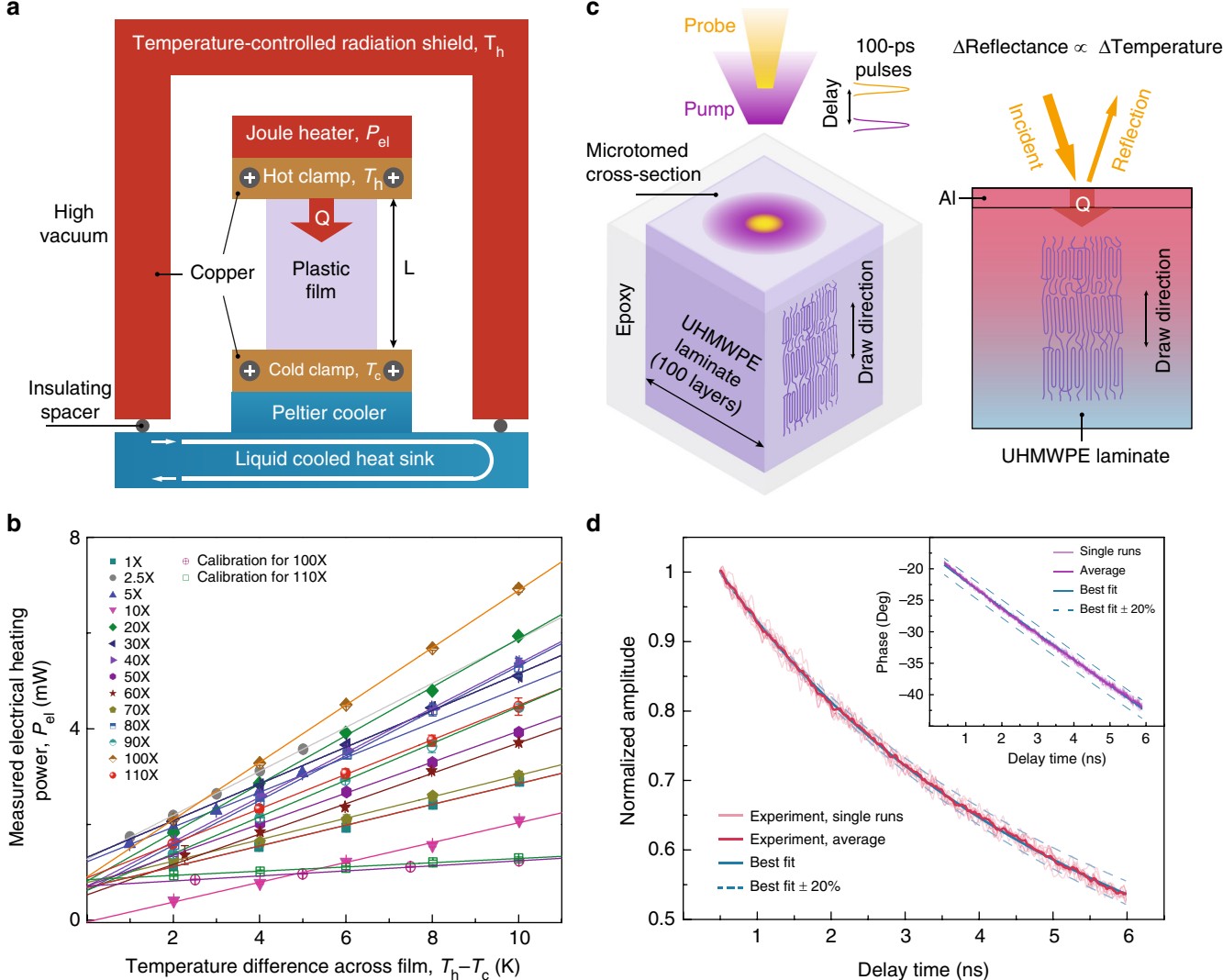

**Fig. 2** Measurement of heat transport along the draw direction of the polymer films. **a** Schematic of the home-built steady-state thermal conductivity measurement system. A small temperature difference ($T_h - T_c$) across a film sample is created and maintained using Joule heating (electrical heating power, $P_{el}$, see Supplementary Note 2 and Supplementary Fig. 1c, d for more details) and thermoelectric cooling inside a high vacuum chamber (Supplementary Fig. 1). **b** Measured electrical heating power ($P_{el}$) as a function of the temperature difference ($T_h - T_c$) across films. The error bars represent the maximum and minimum electrical heater power values measured over the course of 1 min at a sampling rate of 1 Hz. (Supplementary Note 2 and Supplementary Fig. 1b, c). **c** Illustration of the two-color time-domain thermoreflectance measurement scheme. An aluminum-coated UHMWPE laminate is first heated with a 100-fs-wide pump laser pulse (400 nm, purple) and subsequently monitored with a time-delayed low-power probe pulse (800 nm, yellow). The change in aluminum reflectance is proportional to surface temperature variation in the linear regime. **d** Ten individual cooling curves in terms of signal amplitude (light red lines), overlaid with their average (thick red) and the best fit curve (blue solid) that yields a thermal conductivity of 31.9 Wm$^{-1}$K$^{-1}$. Changing the best fit by 20% leads to large discrepancies between the simulated (blue dashed) and measured curves. Inset shows the corresponding phase signals, fitting to which yields a thermal conductivity of 32.8 Wm$^{-1}$K$^{-1}$ (Supplementary Note 3)

small-angle synchrotron X-ray scattering (Fig. 4 and Supplementary Note 4). Wide-angle X-ray scattering (WAXS) measurements were used to determine the crystallite orientation and crystallinity. Comparisons between the as-extruded and drawn films show a clear transition from concentric rings characteristic of polycrystalline samples to short arcs (×10), which eventually become discrete spots (×110), suggesting improved alignment of initially randomly-oriented crystallites upon drawing (Fig. 4b). Specifically, the initially isotropic peaks in the {hk0} group became narrow and oriented along the meridian direction c, indicating that the c-axis (chain direction, Fig. 4a) aligns with the draw direction. The degree of orientation was quantified via an intensity-weighted average over the angle between the c-axis and

the draw direction (Fig. 4a, Supplementary Fig. 9c and Note 4)[32]. The orientation order parameter quickly increases from zero for as-extruded films to nearly saturated value for perfectly aligned crystals at a draw ratio as low as ×2.5 (Fig. 4d). The thermal conductivity of the ×2.5 films (4.5 Wm$^{-1}$ K$^{-1}$) was over 10 times larger than the as-extruded ones (0.38 Wm$^{-1}$ K$^{-1}$, Fig. 3). We therefore expect the excellent alignment of the crystallites to be responsible for the limited thermal conductivity enhancement at very low draw ratio, which is consistent with the conventional strategies to improve the thermal transport in polymers[11].

However, after ×10 draw ratio where the orientation factor nearly saturates, we observed an additional 10-fold thermal conductivity enhancement to the 62 Wm$^{-1}$ K$^{-1}$ (×110), which

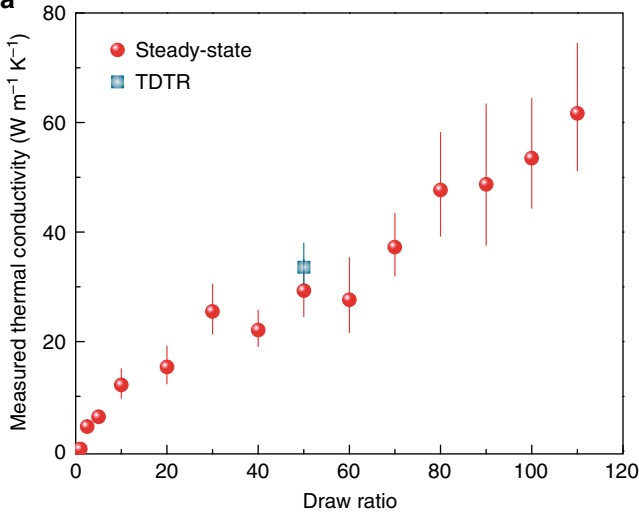

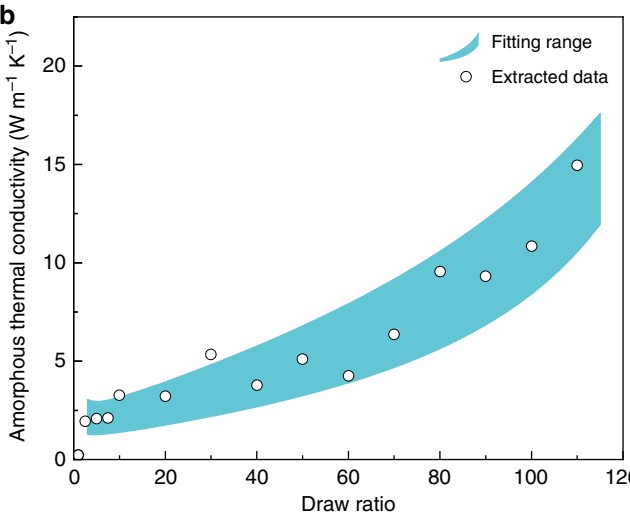

**Fig. 3** Measured and computed thermal conductivities for the polymer films. **a** Measured total thermal conductivity as a function of draw ratio. The red spheres were obtained from the steady-state experiments. A thermal conductivity of 62 W m$^{-1}$ K$^{-1}$ was measured from the ×110 films. The blue square denotes the average of 20 transient thermoreflectance measurements at 3 and 6 MHz modulation. The steady-state error bars take into account the uncertainties in the measurement of the sample geometry, the uncertainty in the estimation of the radiation contribution and the uncertainty in the thermal shunting measurement. See method (thermal conductivity measurements) and Supplementary Note 2 for more details. The TDTR error bar represents the standard deviation (s.d.) over 20 independent measurements at a representative location using both 3 and 6 MHz pump modulation. Fitting to the signal amplitude and phase agreed very well (Supplementary Note 3, Figs. 7 and 8). **b** Extracted amorphous thermal conductivity values based on fitted structural parameters from SAXS analysis. The dots are calculated using the measured total thermal conductivities and modeled crystalline phase thermal conductivity. The shaded region is obtained by fitting the total thermal conductivity with a straight line, and further adding the uncertainties in the determination of structural parameters, thereby giving the estimation of the amorphous thermal conductivity between the upper and lower bound (see more details in Supplementary Note 5)

clearly suggested other enhancement mechanism. We noticed that during the stretching the crystallinity first increased at a high rate at low draw ratios (below ×10) and then steadily grew to over 90% in ×110 films (Fig. 4d, Supplementary Fig. 9f and Note 4).

Contrary to the past work that emphasized on crystallinity-dependence of the thermal conductivity[12], the weak growth rate of crystallinity at high draw ratios is clearly not sufficient to account for the dramatic boost of the thermal conductivity, and there is even no sign of saturation of the conductivity as the draw ratio increases (Fig. 3).

These observations convinced us that unknown factors other than the crystalline phase play the crucial roles especially at high draw ratios. We therefore resorted to the structures of the amorphous region for clues. Quantitative analysis of small-angle X-ray scattering (SAXS) intensity profiles along the draw direction reveals two humps at scattering vectors that differ by a factor of two (Fig. 4e and Supplementary Note 4), indicating a periodic structure with a repeating unit consisting of alternating crystalline and amorphous phases (Fig. 1d, Supplementary Figs 10 and 11)[33]. This picture agrees with the widely-accepted lamella-like structural model for stretched polyethylene[7]. The displacement of the humps toward smaller scattering vectors with increasing draw ratio indicates that the period length grows with drawing. Normalized electron density profiles further reveal the relative lengths of crystalline and amorphous regions in each unit (Fig. 4f inset). Specifically, the amorphous fraction decreases with increasing draw ratio (Fig. 4f), consistent with the increasing trend of crystallinity (Fig. 4d) and film thermal conductivity (Fig. 3a). We have also converted the WAXS data in Fig. 4d into amorphous to superlattice total length ratio and plotted them in Fig. 4f to further illustrates the consistency between the SAXS and WAXS data (see Supplementary Note 4 for more details). However, decreasing the fraction of amorphous region alone cannot account for such observed ultrahigh conductivity, because amorphous phase is simply too thermally resistive[6].

## Discussion

To provide further evidence of the dominant role of amorphous region, a phenomenological one-dimensional thermal transport model is developed (Supplementary Note 5). Based on the structural parameters obtained in WAXS and SAXS, the crystalline and amorphous regions are randomly mixed in the as-extruded films. Upon stretching, aligned fibers consisting of alternating crystalline and amorphous regions are developed in the interior of the film. The average fiber diameter was estimated to be ~10–50 nm nanometers (Fig. 1i, j, Supplementary Figs. 3c and 11c), justifying the use of a one-dimensional model $k = [(1-\eta)/k_c + \eta/k_a]^{-1}$ for the axial thermal conductivity. Here $\eta$ is the amorphous fraction in a periodic unit, and can be fitted from the SAXS analysis (Fig. 4f and Supplementary Note 5), while $k_c$ and $k_a$ are the thermal conductivities of the crystalline and amorphous regions, respectively. The crystalline thermal conductivity $k_c$ depends on the crystallite size due to phonon scatterings at laterally boundaries as well as the length direction. Huang et al.[20] used first-principles to calculate thermal conductivity of 1D polyethylene chain and bulk crystals, and they also discussed size based on diffuse phonon scattering at boundaries. We choose to use their thermal conductivity data for 1D chain as a function of the chain length as values of the crystalline region, since diffuse boundary scattering could be too severe an assumption due to the weak interaction between the crystalline and amorphous phases and possibility continuity of polymer molecules from the crystalline to the amorphous region. Combined with our measured total thermal conductivity, these yield the amorphous thermal conductivity $k_a$ as the draw ratio (Fig. 3b). We do caution that this estimation is subject to uncertainties in $k_c$ and believe that our estimation of the amorphous region thermal conductivity represents a lower bound (Supplementary Note 5). It is clearly seen from Fig. 3b that the

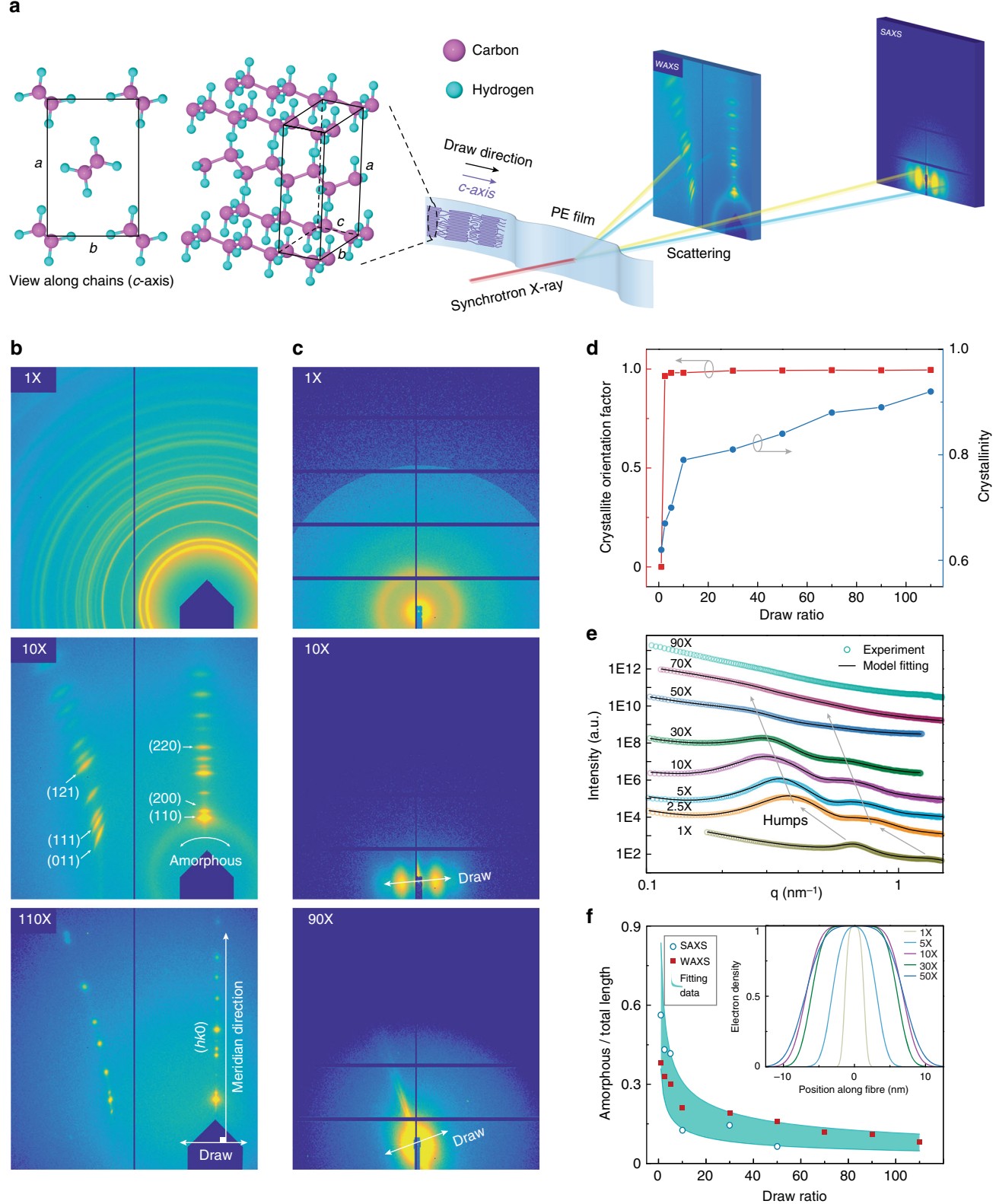

experimentally measured high thermal conductivities at higher draw ratios suggest a high $k_a$ (~5.1 W m⁻¹ K⁻¹ at ×50 and 16.2 W m⁻¹ K⁻¹ at ×110 versus typical 0.3 W m⁻¹ K⁻¹). In other words, the amorphous region after drawing is no longer composed of random disordered chains, but rather has developed some degrees of the orientation order with more extended and

aligned chains. This is also consistent with our experimental observation that the isotropic amorphous diffusing ring gradually disappeared from ×10 to ×110 (WAXS, Fig. 4b and Supplementary Fig. 12), and consistent with the Raman study by Zhu et al. on further stretched Spectra fiber[17]. The extracted high thermal conductivity of the amorphous region with some molecular

**Fig. 4** Structural characterization using synchrotron X-ray scattering. **a** Illustration of the experimental setup and the orthorhombic unit cell of crystalline polyethylene. The incident beam is perpendicular to the drawn direction. The lattice constants were obtained as $a = 7.42$, $b = 4.95$, $c = 2.54$ Å, where c-axis is the chain direction. **b** Wide angle X-ray scattering (WAXS) patterns from the ×1, ×10, and ×110 films. Characteristic Bragg scattering by the {$hk0$} and {$hk1$} plane groups were observed. The {$hk0$} group appears perpendicular to the draw direction. **c** Small angle X-ray scattering (SAXS) patterns from the ×1, ×10, and ×90 films, which clearly show an isotropic-to-anisotropic transition. **d** First-order orientation parameter and the effective crystallinity obtained from WAXS. **e** Scattering intensity linecuts of the SAXS patterns along the draw direction. Two humps appeared at scattering vectors that differ by a factor of two, suggesting a periodic structure with a repeating unit consisting of alternating crystalline and amorphous phases (Supplementary Note 4). The humps moved toward a smaller **q** with increasing draw ratio, indicating an increase in the period length. **f** The fraction of amorphous region in one periodic unit as a function of draw ratio (Supplementary Fig. 14). The blue circles were directly extracted from the SAXS data, while the shaded zone marked the range (±40%) of fitted data which were used in the one-dimensional thermal model (Supplementary Note 5). The red squares recast the crystallinity data in Fig. 4d. Inset is the normalized electron density profile obtained from SAXS analysis (Supplementary Note 4)

orientation is much higher than that of oriented polythiophene fibers grown in a template[19], despite theoretical prediction of higher thermal conductivity of polythiophene than polyethylene in crystal form[34].

In summary, we have developed a scalable manufacturing process for producing polymer films with metal-like thermal conductivity. Unlike conventional approaches focusing on crystalline phase in polymers that can only marginally increase the thermal conductivity, we engineered the none-crystalline chain through disentanglement and alignment and achieved remarkably high thermal conductivity. The past few years have witnessed a surge in the interest of using polymers for thermal management and energy conversion. We believe that the high thermal conductivity achieved in these polymer films, with their unique combination of characteristics (light weight, optical transparency, chemical stability etc.) will play a key role in many existing and unforeseen applications. Of course, polyethylene itself has limitations in the temperature range it can cover. We foresee that further improvement of the thermal conductivity of the persistent amorphous phase will be the key developing the next generation of heat-conducting polymers, in polyethylene and beyond.

## Methods

**Fabrication of thermally conductive polyethylene films**. See Supplementary Note 1 for more details on solution preparation, extrusion, and drawing process. The draw ratios were obtained as the ratio of final to initial film length, with ~20% uncertainty.

**Thermal conductivity measurements**. Direct measurement of the electrical heating power ($P_{el}$) as a function of temperature difference ($T_h - T_c$) across a sample film was performed (Fig. 2a, Supplementary Fig. 1 and Supplementary Tables 1, 2 and Note 2)[28]. Briefly, $T_h$ (303 K, hot clamp temperature) was kept constant via feedback control of $P_{el}$, while $T_c$ (cold clamp temperature) was reduced to create a small temperature difference (up to 10 K) by systematically increasing the thermoelectric cooling power. Multiple measurements of $P_{el}$ were performed at a given temperature difference once the system had reached steady state (Supplementary Note 2). Subsequently, the slope of the linear fit yields, according to Fourier's law and after correction for thermal shunting, the film thermal conductance ($G$) which further gives the film thermal conductivity ($k$) as $k = G \cdot L/A$ (Supplementary Fig. 1b). Here, $L$ and $A$ are respectively the film length and cross-sectional area (Supplementary Tables 1 and 2), which were measured using a suite of tools including a micrometer, optical microscope and profilometer (Supplementary Fig. 2). Special effort was taken to minimize the thermal radiation exchange and to ensure that the reported thermal conductivity is conservative even if any residual radiation exists (Supplementary Note 2). The parasitic radiative heat loss between the sample and the radiation shield is estimated based on the sample emissivity. Thermal shunting was quantified after each experiment by removing the film sample and repeating the measurement (see Supplementary Fig. 1). The reported thermal conductivity is conservative due to the thermal interface resistance between the sample and Cu clamps. The thermal contact resistance was minimized by the thermal paste (Fig. 2a, Supplementary Fig. 1a and Note 2).

For error analysis (parasitic heat losses), we kept the hot clamp and copper radiation shield at the same constant temperature so that parasitic heat losses such as those through the electrical leads to the heater and thermocouples

were effectively kept constant (Supplementary Fig. 1b). Therefore, parasitic heat losses did not affect the slope of $P_{el}$ versus the temperature differential $\Delta T$. We made special effort to minimize the thermal radiation exchange and to ensure that the reported thermal conductivity is conservative even if any residual radiation exists (Supplementary Note 2). Thermal shunting radiation: we kept the thermal conductance values for our samples large enough such that the thermal shunting radiation power (calibration) never exceeded 20% (see Supplementary Figs 1e, f and Note 2). We did calibration measurements without a sample, which directly measured the radiative thermal shunting between the heater and the cold side sample clamp after a sample measurement, and corrected that for parasitic shunting heat loss (Fig. 2b, Supplementary Fig. 1c, f). Finally, we did the measurements on the reference samples (Dyneema, Zylon, and stainless steel, Sn, and Al) and these measurements are in general agreements with literature values.

We also measured the thermal conductivity by time-domain thermoreflectance (Supplementary Note 3). A 100-fs-wide pump laser pulse (~400 nm center wavelength) was used to instantly heat up the surface of an aluminum-coated sample, the cooling of which was then monitored using a probe pulse (800 nm) as a function of delay time between the pulses (Fig. 2c and Supplementary Fig. 6)[31]. Subsequently, the cooling curves were fitted to a standard two-dimensional heat transfer model to get the sample thermal conductivity (Fig. 2c, d and Supplementary Note 3). In order to increase the signal-to-noise ratio, modulated heating was applied by electro-optical modulation of pump power, which resulted in a complex signal with its amplitude and phase recorded by a lock-in amplifier. Both the amplitude and phase signals were used for model fitting. The excellent agreement between amplitude and phase fitting confirmed the measurement reliability (Fig. 2d and Supplementary Fig. 7). Changing the fitted thermal conductivity by 20% led to a large discrepancy between the simulated and measured curves, further indicating good experimental sensitivity (Fig. 2d and Supplementary Fig. 8). The reported value in Fig. 3 was obtained as an average of 20 experiments at 3 and 6 MHz modulation. See Supplementary Note 3 for more details including the specific sample and laser parameters used.

**Structural characterization**. Synchrotron X-ray scattering measurements were used to characterize the film structures at various draw ratios. The experiments were performed at beamline sector 8-ID-E of the Advanced Photon Source, Argonne National Laboratory. See Supplementary Note 4 for details on orientation order parameters and effective crystallinity.

**Thermal conductivity modelling**. We employed a one-dimensional heat transfer model to compute the film thermal conductivity, which depends on the thermal conductivities of the crystalline and amorphous regions, as well as the amorphous fraction ($\eta$) in one periodic unit (amorphous length/period length). The period length was obtained from SAXS structure factor analysis (Supplementary Note 4), while the length of the amorphous region was estimated through the electron density distribution (Fig. 4f, inset and Supplementary Fig. 10) within one period. The experimentally obtained $\eta$ was then fitted to a simple functional form $C_1 * n^{C_2}$, where $n$ denotes draw ratio. Variations of 40% were added to the fitting in order to account for film inhomogeneities and uncertainties involved in the SAXS measurement. The fitted $\eta$ values lead to upper and lower bounds for the computed thermal conductivity at a given draw ratio (Fig. 3b). See Supplementary Note 5 for a detailed description.

## Data availability

The data that support the findings of this study are available from the corresponding authors on reasonable request.

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

## Acknowledgements

The authors acknowledge support from Department of Energy/Office of Energy Efficiency & Renewable Energy/Advanced Manufacturing Program (DOE/EEREAMO) under award number DE-EE0005756 (for the fabrication platform and prior to 2/2016); the MIT Deshpande Center (9/2016-8/2017 for market studies), and U.S. Department of Energy (DOE)–Basic Energy Sciences (Award No. DE-FG02-02ER45977, for fundamental understanding of structure property relationship since 7/2017). This research used resources of the Advanced Photon Source, a U.S. Department of Energy (DOE) Office of Science User Facility operated for the DOE Office of Science by Argonne National Laboratory under Contract No. DE-AC02-06CH11357. The authors thank T. Sanchez and N. Liu from MIT undergraduate research opportunities program for sample preparation; G. Ni for the photograph help; W. Dinatale and L. Wu for SEM discussions; C. Marks and D. Bell for microtome discussions at the Center for Nanoscale Systems, Harvard University; N. Thoppey for his participation in the DOE project; S. Huberman, L. Meroueh and V. Chiloyan for time-domain thermoreflectance discussions; C. Settens (CMSE, MIT), H. Li (Northeastern University), M. Minus (Northeastern University), S. Billinge (Columbia University), and M. Terban (Columbia University) for providing X-ray diffraction testing and discussions; G. Ni, J. Tong, S. Boriskina, T. Cooper, Y. Huang, Q. Song, and L. Weinstein for emittance discussions. S. Huberman, V. Chiloyan, J. Mendoza, L. Meroueh, G. Ni, T. Cooper for the discussion and proofreading, and Jiongzhi Zheng and Professor Baoling Huang (The Hong Kong University of Science and Technology) for original simulation results on thermal conductivity (along the chain direction) as a function of the 1D polyethylene chain[20]. We would like to note that our manuscript has been posted on arXiv: Yanfei Xu et al. Nanostructured polymer films with metal-like thermal conductivity. https://arxiv.org/abs/1708.06416 (2017).

## Author contributions

The materials were fabricated by Y.X. The material thermal properties were characterized by D.K., B.S., Y.X. and J.Z. The material structures were characterized by Y.X. and Z.J. Structural and thermal modelling were performed by Z.J. and J.Z., respectively. Custom-built continuous production platform was built by J.L. J.W., M.L., H.G., X.H., X.L. participated in different phases of this project and contributed to the discussion and understanding of the materials. The manuscript was written by Y.X., B.S. and Z.J. with comments and inputs from all authors. G.C. directed the research.
