## [Peer Review File · Nature Communications]

Reviewers' comments:

Reviewer #1 (Remarks to the Author):

The authors describe a quantitative comparison between thermal conductivity and SAXS/WAXS microstructural characterization experiments for drawn UHMW PE films, which show substantially increased thermal conductivity correlated initially to increased crystallization and later to growth of the crystalline domain length in the drawing direction. Via data analysis they claim that the thermal conductivity of "amorphous" regions separating large crystalline regions becomes unusually high (up to 16 W/m-K) in highly drawn films but still remains the limiting factor preventing even higher overall thermal conductivity.

Up until the second-to-last paragraph (beginning "To provide further evidence of the dominant role of amorphous region..."), I found all of the research claims in the manuscript to be highly original, well-justified, and certainly of interest to the thermal transport community (I am unsure about the appeal to polymer physics community given that the crystallization of PE is fairly well studied). The approach of quantitatively using beam-line scattering/structural characterization to understand the evolution of transport processes is new, and was badly needed for this class of materials. I think this publication will have a high degree of influence on future thinking for these materials and other similar classes of materials.

On the other hand, to the extent that I understood what claim was being made about the nature of the "amorphous" material in the highly drawn samples, I am skeptical. The analysis all seems to rest on that second-to-last paragraph, where there are several critical steps that go into the data analysis which don't look correct to me. The equation

$$1/ktot = (1-\eta)/kc + \eta / kamorphous$$

is being used to extract the value of kamorphous (plotted in figure 3b), meaning that they rely on measured or extracted values of eta (crystalline fraction), kc, and ktot. But this presents difficulties on several levels.

kc cannot be measured (and is a function of length). In essence, the authors are assuming that they know the correct value of kc from ab initio theory (the paper they cite is currently only published in the Arxiv, and I don't have the expertise to comment on its reliability) AND that they know the proper length dependence. Taking into account the inferred length dependence, at around a draw of 100X, the crystalline phase is modeled to have $k \sim 80$ W/m-K (10 W/m-K error bar if figure S15 is correct) which is not all that different from the TOTAL measured thermal conductivity of 60 W/m-K +/- 10 W/m-K. If the $kc = kt$, then obviously $ka = 0$. It seems plausible to me, given the uncertainties in both ka and $ktot$, that ka could be actually close to zero and the conclusion reached in final paragraph and the abstract would be incorrect. This issue is compounded by the fact that the fraction of the amorphous phase is quite small, meaning that any discrepancy between the $ktot$ and kc gets amplified, because of the low sensitivity, into ka .

With the exception of their interpretation of the thermal conductivity of the amorphous phase, I would recommend all other portions of this manuscript for publication as there is some excellent science otherwise in the paper.

Reviewer #2 (Remarks to the Author):

The manuscript by Yanfei Xu, et al. entitled Nanostructured Polymer Films with Metal-like Thermal Conductivity reports a very surprising claim that nanostructured drawn polymer films exhibit thermal conductivities exceeding 30 W/m-K. This claim is augmented by detailed structural characterization and additional thermal analysis to frame a hypothesis that the increase in thermal

conductivity is attributable to a surprisingly high amorphous thermal conductivity contribution. The primary conclusions of this work hinge on the measurement results from a homebuilt steady-state thermal conductivity setup that was previously developed in the group to measure the thermal conductivity of thermoelectric materials and devices (reported in Review of Scientific Instruments, circa 2014, reference numbers 24, 32, and 33 in the main manuscript authored by the second author in this manuscript). The measurement technique used in those previous studies was repurposed for this study.

After careful review of the data, further review of the group's previous publications, and subsequent calculations and analysis, I have very serious concerns that there is systematic error in the thermal conductivity measurement approach used for these samples resulting in an overestimation of the thermal conductivity and therefore has significant consequences to the conclusion reported in this manuscript. In the following, I will describe my observations with the reported data and my reasoning that led me to this position.

First, consider the experimental data in Fig. 2b. As the draw ratio increases the slope becomes more steep, consistent with an increase in thermal conductivity. Upon closer inspection (and linear extrapolation) it is evident that as the draw ratio increases the y-intercept also increases. The y-intercept is non-zero (non-negligibly zero) for all samples (as high as 2 mW or 20% of the fullscale value); it should require zero heating power when the temperature difference is zero. This is evident in the group's previous publications using this measurement technique. However, as the draw ratio increases, a more (positive) amount of electrical power is required to achieve a zero-temperature difference. This should not be the case and I think this is the origin of the systematic error in the measurement.

Next, I tried to understand the origin of this y-intercept as it was not present in the group's previous publications. Insufficient detail is presented in the main manuscript but additional detail is presented in the Supporting Information. Figure S1-b attributes the y-intercept to parasitic heat losses and the text in section 1.2.1 states that the "parasitic heat losses such as through the electrical leads to the heater and the thermocouple were effectively kept constant, which therefore did not affect the slope of P_{el} versus the temperature differential ΔT ." This is the case in the group's previous publications. However, from the data in Fig. 2b, the parasitic heat loss varies with each sample (positively correlated to the draw ratio), so it is evidently not constant and the data counters this statement.

Hence from the data in Fig. 2 and this statement, I then hypothesize that, since the parasitic heat loss systematically increases with increasing draw ratio, there may exist systematic error that causes the slope to systematically increase as well (potentially bringing the results and conclusions into question). I wanted to learn more about how the measurement technique was performed (see schematic in Fig. 2a and photograph in Supporting Information Fig. S1), so I carefully reread the group's previous publications and looked closely at the Supporting Information. A detail in Fig. S1 caught my attention; specifically, I note that the clamps are over 2 cm and serve as a scale bar. Previous measurements using this homebuilt apparatus were used to measure thermoelectric elements, which were nominally 1.64 mm cubes (Ref. 24), resulting in a thermal conductivity of ~ 1.4 W/m-K. Thus, the thermoelectric samples being measured had a thermal conductance around ~ 2.3 mW/K. This is a nominal value for which I took the measurement technique to be valid; values significantly above or below this value may be questionable. This image in Fig. S1 shows a translucent film sample whose length is comparable to the clamps size (2 cm). Assuming a thermal conductivity of 30 W/m-K and a nominal cross sectional area (as reported in Table S1) of 0.02 mm², the conductance is 0.03 mW/K with a 2 cm length. This is nearly two orders of magnitude smaller than the nominal value and may therefore be outside the technique's measurement capabilities. Table S1 reports the lengths were on the order of 1-2 mm (Are the authors sure the lengths were mm's, not cm's as shown in the photograph? One length is even reported to be as small as 650 μ m for one of the highest claimed thermal conductivities.); this makes the conductance 0.3 mW/K (nearly one order of magnitude smaller than the nominal

value). Assigning a linearized radiation conductance, G_{rad} , [$Q_{\text{rad}} = \sigma (\epsilon A) (T_h^2 + T_c^2) (T_h + T_c) (T_h - T_c) = G_{\text{rad}} (T_h - T_c)$] with an (ϵA) ranging between 5 cm^2 and 50 cm^2 to cover possible emissivity and view factor area combination and $T_h = 300 \text{ K}$ and $T_c = 298 \text{ K}$ (note: a modest 2 degree temperature difference), I estimate that the radiation conductance to be between 3 mW/K and 30 mW/K , which is one to three orders of magnitude greater than the conductance through the polymer film. As a result of these calculations, I have serious concerns regarding the appropriateness of this technique to measure thin-film samples unless the thermal conductivity exceeds 300 W/m-K or lengths are below 100 um . There is a statement that says "Special effort was taken to minimize the thermal radiation exchange and ensure that the reported thermal conductivity is conservative even if any residual radiation exists." However, I do not believe sufficient effort was taken to minimize radiation from the photographs of the setup and find that the radiation component could be much greater than the conductance through the film.

Furthermore, I also note that in Fig. S1, the sample extended above the hot junction heat clamp (potentially in contact with the radiation shield). The y-intercept/parasitic heat loss data suggests that heat from the hot junction systematical transfers (or is lost) to either (i) the radiation shield or (ii) the cold clamp, increasing in magnitude with the sample order of the increasing draw ratio. I'd also like to note that I think the polymer films become more transparent (to IR) as the draw ratio increases, but data is not presented or available to support this conjecture; this could be the reason why the systematic error and parasitic heat loss trends with the draw ratio if radiation dominates. Alternatively, for draw ratios 50 through 100, I notice a systematic decrease in the length from 2.25 mm to 0.65 mm , and a systematic decrease in cross sectional area from 0.021 mm^2 to 0.0069 mm^2 , which are the range of draw ratios where the reported thermal conductivity monotonically exceed 30 W/m-K and could be responsible for the systematic error.

Nevertheless, in the case of (i), this means that the radiation shield and the hot junction are not at the same temperature (as controlled by the feedback loop; maybe they are slightly off by one of two degrees spatially). In this case, the radiation shield would be at a lower temperature than the hot junction heat clamp. Looking at the image in Fig. S1, I suspect it could be that the insulating spacers (Fig. 2a) may not be insulating enough to prevent the shield from being cooled (maintaining a uniform temperature equal to the hot clamp temperature everywhere) or the Peltier cooler is radiatively cooling the radiation shields or hot clamp (which I suspect as the degree of parasitic heat loss correlates with a decrease in length indicative of a view factor change). Since the hot junction temperature is fixed and the cold clamp temperature is lowered, there would then be a systematic over approximation of the heat transferred from the joule heater. Hence, the measured value would increase as the temperature difference increases (thereby artificially increasing the slope) and the thermal conductivity of the sample would be overestimated. In the case of radiatively cooling the radiation shield. This overestimation would be non-linear with the temperature difference over a wide range but linear over the small temperature differences in this study and would systematically scale with the increase in thermal conductivity of the sample.

In the case of (ii), there would exist a bypass heat pathway (potentially radiation) between the hot clamp and cold clamp. As the temperature difference between these clamps increases so would the amount of bypass heat. Hence, the measured value would increase as the temperature difference increases (thereby artificially increasing the slope) and the thermal conductivity of the sample would be overestimated.

Thus, in both cases the thermal conductivity of the sample would be overestimated. However, the authors performed a single TDTR measurement and found that the thermal conductivity was $\sim 32 \text{ W/m-K}$. While TDTR is a challenging technique to perform on these samples, why not validate the claim of 62 W/m-K (or essentially anything greater than $\sim 30 \text{ W/m-K}$) with TDTR? This seemed very odd to me and for such a claim as "Nanostructured Polymer Films with Metal-Like Thermal Conductivity"; that validation should be performed on all samples exceeding 30 W/m-K . Furthermore, why not validate this with more standard thin-film techniques like 3-omega rather

than the hassle and potential error with microtoming? In-plane and through-plane thermal conductivity can be measured by varying heater line widths using 3-omega (I can point to Rev. Sci. Instr. manuscripts that demonstrate this if the authors are unfamiliar). Or using TDTR or FDTR, in-plane thermal conductivity can be measured using the beam offset technique (I can point to Rev. Sci. Instr. manuscripts that demonstrate this if the authors are unfamiliar). Even transient grading or laser flash could validate this claim. It seems very odd that such claims are made hinging on a homebuilt (non-standard) technique that is utilized outside its measurement range where radiation potentially dominates, when alternative options are available.

In summary, I have very serious concerns with the thermal conductivity measurements as I suspect that radiation is the dominant heat loss mechanism (not conductance through the polymer film) and is the source of the systematic error (parasitic heat loss and y-intercept). Thus, I think the reported value of thermal conductivity is overestimated. This then brings into question the subsequent analysis and conclusions.

To improve the manuscript, to refute these observations, and to convince this reviewer of the validity of the measurements and conclusions of this work, please perform all of the following:

- 1) Share the data of the control experiments where no sample is present where the clamp heater is held at various heights above the cold clamp to demonstrate radiation is not significant.
- 2) Perform a thermal conductivity measurement of a copper foil using the homebuilt steady-state system as a function of length, reporting the thickness and width (including a photograph).
- 3) Perform measurements on standards of (opaque) films ranging from 30 W/m-K to 60 W/m-K with the homebuilt technique (without systematic error).
- 4) Explain the systematic error and increase of the large (~2 mW of 10 mW) parasitic heat loss that follows the sample order of increasing draw ratio (or decreasing length, or increasing transparency).
- 5) Perform TDTR (beam offset), 3-omega (multiple heater line width), transient grading, or laser flash measurements on the samples with draw ratios of 30, 40, 60, 70, 80, 90, 100, and 110. Try for anisotropic measurements using beam offset or multiple heater line widths.

And

- 6) Redo amorphous thermal conductivity calculations based on this data (Figs. 3b and S15).

Reviewer #3 (Remarks to the Author):

This work provides important data and understandings regarding thermal conductivity in drawn polyethylene films. The authors find that the films consist of nanofibers with alternating crystalline and amorphous regions, which is consistent with lamellar structure expected for drawn polyethylene. The thermal conductivities of these two regions are not measured directly. The amorphous region's thermal conductivity is derived from fitting experimental data to a 1D thermal model in which the crystalline region is assigned an assumed value based on a first-principles calculation in another work. The authors find that the derived thermal conductivity of the amorphous phase increases with draw ratio, and their overall finding is that thermal conductivity of the amorphous phase is the bottleneck to film thermal conductivity.

In general, I view the topic area of heat conduction in polymers to be an important one, both scientifically and technologically. Furthermore, the structural and thermal data provided by this work will be useful to other researchers studying thermal conductivity in semicrystalline polymers. However, for this particular work I consider general finding that the amorphous areas surrounding lamellar crystals are the primary bottleneck to thermal conductivity to be a reasonably expected result and not of sufficiently broad interest.

Revision Report for

Nanostructured Polymer Films with Metal-like Thermal Conductivity

*Yanfei Xu, Daniel Kraemer, Bai Song, Zhang Jiang, Jiawei Zhou, James Loomis, Jianjian Wang, Mingda Li, Hadi Ghasemi, Xiaopeng Huang, Xiaobo Li and Gang Chen**

e-mail: gchen2@mit.edu

We thank the reviewers for their careful reading and constructive criticism of the manuscript. Our detailed responses are documented below.

Reviewer #1 (Remarks to the Author):

The authors describe a quantitative comparison between thermal conductivity and SAXS/WAXS microstructural characterization experiments for drawn UHMW PE films, which show substantially increased thermal conductivity correlated initially to increased crystallization and later to growth of the crystalline domain length in the drawing direction. Via data analysis they claim that the thermal conductivity of “amorphous” regions separating large crystalline regions becomes unusually high (up to 16 W/m-K) in highly drawn films but still remains the limiting factor preventing even higher overall thermal conductivity.

Up until the second-to-last paragraph (beginning “To provide further evidence of the dominant role of amorphous region...”), I found all of the research claims in the manuscript to be highly original, well-justified, and certainly of interest to the thermal transport community (I am unsure about the appeal to polymer physics community given that the crystallization of PE is fairly well studied). The approach of quantitatively using beam-line scattering/structural characterization to understand the evolution of transport processes is new, and was badly needed for this class of materials. I think this publication will have a high degree of influence on future thinking for these materials and other similar classes of materials.

On the other hand, to the extent that I understood what claim was being made about the nature of the “amorphous” material in the highly drawn samples, I am skeptical. The analysis all seems to rest on that second-to-last paragraph, where there are several critical steps that go into the data analysis which don’t look correct to me. The equation

$$1/k_{tot} = (1-\eta)/k_c + \eta / k_{amorphous}$$

is being used to extract the value of $k_{amorphous}$ (plotted in figure 3b), meaning that they rely on measured or extracted values of η (crystalline fraction), k_c , and k_{tot} . But this presents difficulties on several levels.

k_c cannot be measured (and is a function of length). In essence, the authors are assuming that they know the correct value of k_c from ab initio theory (the paper they cite is currently only published in the Arxiv, and I don't have the expertise to comment on its reliability) AND that they know the proper length dependence. Taking into account the inferred length dependence, at around a draw of 100X, the crystalline phase is modeled to have $k \sim 80$ W/m-K (10 W/m-K error bar if figure S15 is correct) which is not all that different from the TOTAL measured thermal conductivity of 60 W/m-K ± 10 W/m-K. If $k_c = k_t$, then obviously $k_a = 0$. It seems plausible to me, given the uncertainties in both k_a and k_{tot} , that k_a could be actually close to zero and the conclusion reached in final paragraph and the abstract would be incorrect. This issue is compounded by the fact that the fraction of the amorphous phase is quite small, meaning that any discrepancy between the k_{tot} and k_c gets amplified, because of the low sensitivity, into k_a .

With the exception of their interpretation of the thermal conductivity of the amorphous phase, I would recommend all other portions of this manuscript for publication as there is some excellent science otherwise in the paper.

Response: We thank the reviewer for acknowledging the significance of our work, and especially the structural characterization we applied for understanding the details behind the large thermal conductivity (“highly original, well-justified, and certainly of interest to the thermal transport community”, “The approach of quantitatively using beam-line scattering/structural characterization ... is new, and was badly needed for this class of materials”). The reviewer’s major concern is that the amorphous thermal conductivity does not need to be high to explain the measured total thermal conductivity. We agree with the reviewer that there are uncertainties involved in determining the amorphous phase thermal conductivity, but believe that we have carefully examined these uncertainties to draw solid conclusions. As the reviewer pointed out, the estimated thermal conductivity depends on the amorphous fraction and the crystalline state thermal conductivity we used. In the following, we provide a careful sensitivity analysis regarding these two parameters, and show that in any condition consistent with our quantitative structural characterization, the thermal conductivity of the drawn amorphous phase would be significantly larger than bulk polyethylene. The discussion on the sensitivity to experimental parameters has been added as a new section into the Supplementary Information in the revised version (SI, Section 1.4.2).

First, we would like to point out that *ab initio* simulation of phonon thermal conductivity based on DFT has matured to a level of good reliability¹⁻³. The Kaviani group which published the above-cited arXiv paper (Ref. 18 in the manuscript) has published many works previously using this approach. The corresponding author's own group has also done lots of work using this approach and has good confidence on its accuracy. Second, the corresponding author's group also conducted molecular dynamics simulations before and found that single polymer chain could have higher, even divergent thermal conductivity⁴. This effect, which is due to phonon correlation, cannot be captured by the BTE-based transport calculation. In this sense, Ref. 18 may have underestimated the thermal conductivity of ideal polyethylene crystals. Our confidence on Ref. 18 was based on such experience. Figure S15b is calculated using Matthiessen's rule based on the spectral phonon MFP distribution, as reported in Ref. 18 (Fig. 6b in Ref. 18). Past work has proven that these approaches are reasonable. Since our fiber geometry is even simpler than past polycrystalline and nanostructured samples, the application of this model to our current problem is well justified.

Now we provide a sensitivity analysis for the relevant parameters, using 50× drawn sample as an example. We first evaluate the sensitivity of the estimated amorphous thermal conductivity to the value of the crystalline state thermal conductivity. In Fig. R1a, we plot the variation of the estimated thermal conductivity (k_a) with the assumed crystalline state thermal conductivity (k_c). The value of k_c is determined based on the computed polyethylene thermal conductivity together with measured crystallite size, so that size effect can be properly taken into account (Fig. S15). At 50× we obtained $k_c \approx 70$ W/m-K, corresponding to $k_a \approx 5$ W/m-K. Although there are uncertainties involved in determining the crystallite size, it is clear from Fig. R1a that k_a is very insensitive to k_c , and for any value of k_c , k_a is always larger than ~ 4 W/m-K, significantly larger than its bulk case. This is because, as long as there is a certain fraction of the amorphous phase in series with the crystalline phase, the amorphous thermal conductivity has to be sufficiently large to ensure a large total thermal conductance. As long as $k_c \gg k_a$, the uncertainties in k_c do not affect our conclusion.

Of course, if the fraction of the amorphous region is zero, as the reviewer suggested, a crystalline thermal conductivity close to our measured total thermal conductivity could then well explain the data without invoking large amorphous phase thermal conductivity. However, we believe our structural characterization has shown that the fraction of the amorphous phase is not zero. We thank the reviewer again for acknowledging the importance of our beamline-based quantitative characterization. Specifically, we obtain our amorphous fraction via the normalized electron density profile from the small-angle X-ray scattering (SAXS) measurement (Fig. 4f inset and Fig. S10). The normalized electron density continuously varies from 1 to 0, indicating the presence of two phases in the drawn sample. The regions with value close to zero represent the amorphous phase, while the remaining ones (increasing length as the draw ratio, and include the transition region as well) are regarded as the crystalline region. The amorphous fraction is

estimated by dividing the length of the amorphous phase by the period length obtained from SAXS (SI, Section 1.4). Using the 50 \times sample as an example, here we examine the sensitivity of k_a to the amorphous fraction parameter (η), as shown in Fig. R1b. At 50 \times , our estimated amorphous fraction is $\eta \approx 11\%$ (Fig. 4f), corresponding to $k_a \approx 5$ W/m-K. This amorphous fraction is consistent with the crystallinity of the 50 \times samples (84%, Fig. 4d) measured using wide-angle X-ray scattering (WAXS). We have estimated the amorphous fraction based on SAXS rather than crystallinity because we believe SAXS provides a more quantitative measure. Nonetheless, if we instead inferred the amorphous fraction based on crystallinity to give $\eta \approx 16\%$, k_a will be even higher (~ 7 W/m-K, Fig. R1b). We are aware of the uncertainty involved in determining the amorphous fraction and have taken this into account, shown by the shaded area in Fig. 4f. In case the actual amorphous fraction is 50% less than the current value ($\eta \approx 6\%$), k_a will be ~ 2.5 W/m-K (Fig. R1b), still significantly larger than bulk polyethylene. Therefore, we have shown that within the possible range of the amorphous fraction η , k_a could vary but is always significantly larger than its bulk value.

Figure R1. Dependence of estimated amorphous phase thermal conductivity on **a**, crystalline state thermal conductivity, and **b**, amorphous fraction at 50 \times draw ratio. The red point corresponds to the actual case while the blue line indicates the variation if corresponding parameters are changed.

Last but not least, we note that the structure information (Fig. 4f inset, Fig. S11, Fig. S15a, and SI Section 1.2) are only directly extracted from SAXS up to 50 \times draw ratio, while for higher draw ratios the structure information (amorphous fraction, crystallite size) is extrapolated, which could lead to further uncertainty. However, the WAXS data in Fig.4d also support this extrapolation. Based on the above discussions, we think the increasing thermal conductivity in the amorphous phase is a general trend and should continue as we draw the sample beyond 50 \times . This general notion is important for understanding the limiting factor in the thermal transport of polymers.

Response to reviewer #2:

The manuscript by Yanfei Xu, et al. entitled Nanostructured Polymer Films with Metal-like Thermal Conductivity reports a very surprising claim that nanostructured drawn polymer films exhibit thermal conductivities exceeding 30 W/m-K. This claim is augmented by detailed structural characterization and additional thermal analysis to frame a hypothesis that the increase in thermal conductivity is attributable to a surprisingly high amorphous thermal conductivity contribution. The primary conclusions of this work hinge on the measurement results from a homebuilt steady-state thermal conductivity setup that was previously developed in the group to measure the thermal conductivity of thermoelectric materials and devices (reported in Review of Scientific Instruments, circa 2014, reference numbers 24, 32, and 33 in the main manuscript authored by the second author in this manuscript). The measurement technique used in those previous studies was repurposed for this study.

After careful review of the data, further review of the group's previous publications, and subsequent calculations and analysis, I have very serious concerns that there is systematic error in the thermal conductivity measurement approach used for these samples resulting in an overestimation of the thermal conductivity and therefore has significant consequences to the conclusion reported in this manuscript. In the following, I will describe my observations with the reported data and my reasoning that led me to this position.

Response: We greatly thank the reviewer for taking the time to conduct such a thorough review of our manuscript, and for acknowledging the “surprising” aspect of our results. To begin with, we apologize for some confusions and misunderstandings caused by two of our figures (Fig. 2b and Fig. S1), which we did not explain more clearly and have led to the reviewer's concern about the accuracy of the data. Below we provide a general discussion and point-by-point responses to the reviewer's comments.

The key message here is that Fig. 2b does not show the directly measured electrical heating power P_{el} (raw data) in our experiments, rather, it shows a geometrically scaled electrical heating power (scaled to the geometry of a 50× sample, $P_{els} = P_{el} \cdot (A/L)_{50\times} / (A/L)$, as previously indicated in the caption of Fig. 2). We have added the directly measured electrical heating power P_{el} in our revised Supporting Information (SI) as Fig. S1c-d, which is the same as Fig. R2a-b below.

The scaled data in Fig. 2b shows how the P_{el} vs temperature difference data would look like if the geometry from sample to sample would stay the same. We plotted Fig. 2b this way to show that larger thermal conductivity sample having a larger slope makes comparison between different samples' thermal conductivities more apparent. This data processing leads to an unrealistic increase in y-intercept with sample conductivity which does not happen in the raw data (Fig.

R2a-b). The y-intercept in Fig. R2a-b is randomly different from sample measurement to sample measurement which is expected from possible changing parasitic heat losses due to the sample mounting and slight changes in surface properties of clamps and surrounding over time.

Experimentally, we note that we took great care to reduce any systematic errors. For example, we kept the hot clamp and the copper radiation shield at the same constant temperature, so that parasitic heat losses such as those through the electrical leads to the heater and thermocouples were effectively kept constant, which therefore did not affect the slope of P_{el} versus the temperature differential ΔT (SI Section 1.2.1 Steady-state method). Special effort was taken to minimize the thermal radiation exchange and to ensure that the reported thermal conductivity is conservative even if any residual radiation exists. Further, we optimized the length and cross-sectional area of each sample according to the expected thermal conductivity as much as we could to ensure similar thermal conductance for all samples including the Dyneema⁵, Zylon⁵, and stainless steel⁶ references (Fig. R2c, we have also added this data in Fig. S1e and Table S1), and to ensure that the parasitic heat losses are small and essentially constant from sample to sample. The sample thermal conductance measurement is minimally affected as long as the signal is dominated by the conducted heat through the sample. Moreover, we performed calibration runs after sample measurement by cutting the film and measuring the radiative thermal shunting directly (Fig. R2a-b). Finally, our measurements of the reference samples (Dyneema⁵, Zylon⁵, and stainless steel⁶) are in good agreement with literature values^{5,6}, giving us good confidence in all our measurements.

Figure R2. a Directly measured electrical heating power (P_{el}) for representative samples, including the polyethylene films, the reference samples (Dyneema⁵, Zylon⁵, and stainless steel⁶), and the calibrations without films. **b** Directly measured electrical heating power P_{el} for all the polyethylene films and representative calibrations. **c** Measured thermal conductance of all the polyethylene films and reference samples (Dyneema⁵, Zylon⁵, and stainless steel⁶). Inset is the measured thermal conductance of reference samples. Error bars indicate one standard deviation.

Comment (1):

First, consider the experimental data in Fig. 2b. As the draw ratio increases the slope becomes more steep, consistent with an increase in thermal conductivity. Upon closer inspection (and linear extrapolation) it is evident that as the draw ratio increases the y-intercept also increases. The y-intercept is non-zero (non-negligibly zero) for all samples (as high as 2 mW or 20% of the

fullscale value); it should require zero heating power when the temperature difference is zero. This is evident in the group's previous publications using this measurement technique. However, as the draw ratio increases, a more (positive) amount of electrical power is required to achieve a zero-temperature difference. This should not be the case and I think this is the origin of the systematic error in the measurement.

Response (1): As explained above, Fig. 2b in the original manuscript shows the geometrically scaled electrical heating power input, which means that the shown data (10×, 90×, 110×, Dyneema and S. Steel 304) are scaled to the geometry of a 50× film ($P_{els} = P_{el} \cdot (A/L)_{50\times} / (A/L)$). The data in Fig. 2b shows what slope would be expected if the dimensions of the samples had been the same for all which was not the case. In fact, we did the exact opposite. We adjusted the lengths and cross-sectional areas of the samples such that we could keep a more or less constant thermal conductance for all samples including the reference samples (Dyneema⁵, Zylon⁵, and stainless steel⁶). By doing this we ensure that parasitic heat losses are similar in all measurement (Fig. R2c, the same as Fig S1e and Table S1). The measured thermal conductivity of the reference samples (Dyneema⁵, Zylon⁵, and stainless steel⁶) showed good agreement with literature^{5,6}, which gives us confidence in our measurements of the polyethylene films.

If the y-intercept of the **actual** measured electrical heating power P_{el} (raw data in Fig. R2a-b) would show such a strong dependence on the sample thermal conductivity, the reviewer's conclusion would be absolutely correct. In fact, our raw data shows expected random y-intercepts due to random small changes in parasitic heat losses every time a new sample is attached (to avoid confusion, we have revised Fig. 2b y-axis — geometrically scaled heating power, and explained $P_{els} = P_{el} \cdot [(A/L)_{50X} / (A/L)]$ in the caption). And it is those heat losses and possible small differences in the temperatures between the surrounding and the heater clamp that can lead to a (from sample to sample random) non-zero power input at a measured zero temperature difference. However, as the reviewer acknowledges, the strength of this method is that those random parasitic heat losses do not affect the measurement as long as they are constant during one sample measurement. And the fact, that the y-intercepts are random (see raw data Fig. R2a-b) is a reasonable proof that that is the case. Furthermore, we also added calibration measurements that were taken after sample measurements and the sample was cut to eliminate the heat conduction through the sample (Fig. R3 as an example). This allowed us to directly measure the radiative thermal shunting between the heater and the cold side sample clamp after a sample measurement. The thermal conductance values for our samples were always large enough such that the thermal shunting radiation power never exceeded 25% (Fig. R3) of the heat conducted through the sample. Nevertheless, as stated in the manuscript and SI, we corrected for that parasitic shunting heat loss.

Figure R3. Representative heating power for the 100× and 110× samples, and the corresponding calibration curves without the samples. All error bars indicate one standard deviation.

Comment (2):

Next, I tried to understand the origin of this y-intercept as it was not present in the group’s previous publications. Insufficient detail is presented in the main manuscript but additional detail is presented in the Supporting Information. Figure S1-b attributes the y-intercept to parasitic heat losses and the text in section 1.2.1 states that the “parasitic heat losses such as through the electrical leads to the heater and the thermocouple were effectively kept constant, which therefore did not affect the slope of P_{el} versus the temperature differential ΔT .” This is the case in the group’s previous publications. However, from the data in Fig. 2b, the parasitic heat loss varies with each sample (positively correlated to the draw ratio), so it is evidently not constant and the data counters this statement.

Response (2): We hope our discussions above clarified this question. We added two plots with the directly measured heating electrical power P_{el} (raw data) in the SI (Fig. S1c-d).

Comment (3):

Hence from the data in Fig. 2 and this statement, I then hypothesize that, since the parasitic heat loss systematically increases with increasing draw ratio, there may exist systematic error that causes the slope to systematically increase as well (potentially bringing the results and conclusions into question).

Response (3): There is no large systematic error in our measurement. The largest source of systematic error originates from our sample dimension measurements which are included in the error bars.

Comment (4):

I wanted to learn more about how the measurement technique was performed (see schematic in Fig. 2a and photograph in Supporting Information Fig. S1), so I carefully reread the groups previous publications and looked closely at the Supporting Information. A detail in Fig. S1 caught my attention; specifically, I note that the clamps are over 2 cm and serve as a scale bar. Previous measurements using this homebuilt apparatus were used to measure thermoelectric elements, which were nominally 1.64 mm cubes (Ref. 24), resulting in a thermal conductivity of ~ 1.4 W/m-K. Thus, the thermoelectric samples being measured had a thermal conductance around ~ 2.3 mW/K. This is a nominal value for which I took the measurement technique to be valid; values significantly above or below this value may be questionable.

Response (4): We respectfully disagree with this comment. The lowest measurable conductance value is determined by

- (1) parasitic radiation heat transfer from the sample to surrounding: The effect of sample radiation explained in the SI and calculations show that radiation heat load from the surrounding onto the sample was less than 25% compared to the conducted heat. Also, we would like to point out that this sample radiation heat transfer leads to an underestimate in the measured thermal conductance since the temperature of the surrounding is kept at the heater clamp temperature. Radiation from the shield to sample adds heat flow through the film that is not counted in thermal conductivity calculation. Also, the thermal conductivities of several reference samples (including stainless steel⁶) with similar conductance values (as the polymer samples: Dyneema⁵ and Zylon⁵) were measured and showed good agreement with existing literature.
- (2) radiative thermal shunting loss: we performed calibration measurements to quantify and correct for that thermal shunting heat loss. We also made sure that the conductance values of our samples were large enough such that the radiative thermal shunting heat transfer between the heat and cold side clamp never exceeded 25% compared to the conducted heat through the sample to ensure a large enough signal-to-noise ratio (see Fig.S2d) and is less than 10% for most samples, especially at high draw ratio. We also correct for that thermal shunting heat flow when determining the sample thermal conductivity.
- (3) the fluctuations of the parasitic heat losses from the heater clamp due to uncaptured fluctuating temperatures and changing thermal resistances during the measurement: those fluctuations would show up as deviations from linearity in the measured electrical heating power P_{el} vs temperature difference data which is not the case. All our sample measurements and calibration measurements showed a linear behavior (Fig. R2a-b).

Comment (5):

This image in Fig. S1 shows a translucent film sample whose length is comparable to the clamps size (2 cm). Assuming a thermal conductivity of 30 W/m-K and a nominal cross sectional area (as reported in Table S1) of 0.02 mm², the conductance is 0.03 mW/K with a 2 cm length. This is nearly two orders of magnitude smaller than the nominal value and may therefore be outside the technique's measurement capabilities. Table S1 reports the lengths were on the order of 1-2 mm (Are the authors sure the lengths were mm's, not cm's as shown in the photograph? One length is even reported to be as small as 650 um for one of the highest claimed thermal conductivities.); this makes the conductance 0.3 mW/K (nearly one order of magnitude smaller than the nominal value). Assigning a linearized radiation conductance, G_{rad} , [$Q_{rad} = \sigma * (\epsilon * A) * (T_h^2 + T_c^2) * (T_h + T_c) * (T_h - T_c) = G_{rad} * (T_h - T_c)$] with an ($\epsilon * A$) ranging between 5 cm² and 50 cm² to cover possible emissivity and view factor area combination and $T_h = 300$ K and $T_c = 298$ K (note: a modest 2 degree temperature difference), I estimate that the radiation conductance to be between 3 mW/K and 30 mW/K, which is one to three orders of magnitude greater than the conductance through the polymer film. As a result of these calculations, I have serious concerns regarding the appropriateness of this technique to measure thin-film samples unless the thermal conductivity exceeds 300 W/m-K or lengths are below 100 um. There is a statement that says "Special effort was taken to minimize the thermal radiation exchange and ensure that the reported thermal conductivity is conservative even if any residual radiation exists." However, I do not believe sufficient effort was taken to minimize radiation from the photographs of the setup and find that the radiation component could be much greater than the conductance through the film.

Response (5): We appreciate the reviewer's thoroughness in checking. The image in the original Fig S1a serves mostly an illustrative purpose to clearly show our setup beyond what the schematic in Fig. 2a can convey. As the reviewer noted, the actual dimensions of all samples are given in Table S1. We have never used any cm-long samples. Our samples were, in fact, in some cases as short as 1 mm depending on the cross-sectional area and the roughly expected thermal conductivity value of the sample to ensure similar conductance values for all samples. We have added in Fig. S1 an image more representative of the real experiments on the polyethylene films (70 ×). It seems that the reviewer has based his/her conclusions on the wrong sample geometry. Regarding the reviewer's concern that this measurement technique is not suitable for measuring our sample conductance values, we respectfully disagree. We accurately measured the thermal conductivity of several reference samples (Dyneema⁵, Zylon⁵, and stainless steel⁶) with conductance values similar to our polyethylene samples. So, we feel very confident that our experimental method and setup have the capability to measure the thermal conductivity of our polymer samples.

As explained in the SI, the surrounding temperature is maintained constant at the temperature of the heater clamp – thus, the suspended sample is exposed to a radiative heat load from the

surrounding potentially reducing the required electrical heat input to the heater clamp. If there is a significant radiation heat load onto the sample that would lead to a measured heat flow that is lower than what would be expected for a sample with a certain thermal conductivity. Consequently, the sample radiation effect leads to an underestimate in the sample's thermal conductivity. However, our successful reference sample measurements (Dyneema⁵, Zylon⁵, and stainless steel⁶) support the accuracy of our sample measurements. We also theoretically analyzed and discussed the effect of sample radiation in great detail in the SI. The ratio between sample radiation and sample heat conduction varies between 1-25%. If we take for example our polymer sample with a draw ratio of 110×, the suspended sample length was 1 mm and the cross-sectional area was 0.00525 mm² (3.5 mm × 0.0015 mm) and the measured thermal conductance was 0.358 mW/K. Using the reviewer's radiation analysis (more accurate analysis is given in the SI) the radiation conductance, Grad, is 0.00841mW/K (assuming an emittance of 0.2) which is less than 2.4% relative to the measured thermal conductance of the sample.

Comment (6):

Furthermore, I also note that in Fig. S1, the sample extended above the hot junction heat clamp (potentially in contact with the radiation shield). The y-intercept/parasitic heat loss data suggests that heat from the hot junction systematical transfers (or is lost) to either (i) the radiation shield or (ii) the cold clamp, increasing in magnitude with the sample order of the increasing draw ratio. I'd also like to note that I think the polymer films become more transparent (to IR) as the draw ratio increases, but data is not presented or available to support this conjecture; this could be the reason why the systematic error and parasitic heat loss trends with the draw ratio if radiation dominates. Alternatively, for draw ratios 50 through 100, I notice a systematic decrease in the length from 2.25 mm to 0.65 mm, and a systematic decrease in cross sectional area from 0.021 mm² to 0.0069 mm², which are the range of draw ratios where the reported thermal conductivity monotonically exceed 30 W/m-K and could be responsible for the systematic error.

Response (6): As explained in response (1) the y-intercept in Fig. 2b is meaningless due to the geometric scaling. Therefore, any further conclusions drawn by the reviewer are misguided. The monotonically increasing trend in the y-intercept is due to the geometrically scaling of the raw data NOT due to an increasing systematic error.

For the measurement to work with highest accuracy it is important that all heater wires (voltage, current and thermocouple leads) are thermally grounded to the temperature-controlled guard heater/radiation shield to reduce the parasitic heater heat loss (for high signal-to-noise) and to maintain the parasitic heat loss constant during a sample measurement. Even though all our final measured samples did not extend much beyond the heater clamp, it would not matter as long as the sample is thermally grounded to the guard heater/radiation shield. We have paid special attention to this point, and performed analysis as shown in Fig. S4.

The reviewer's expectation that thinner films are more transparent is correct, but that means less radiation heat transfer. We used literature values on emittance versus draw ratio (Fig.S2c). Our own test gives us smaller emittance values. Hence, we believe that literature values provide a conservative estimate that is sufficient for establishing the accuracy of our experiment.

Comment (7):

Nevertheless, in the case of (i), this means that the radiation shield and the hot junction are not at the same temperature (as controlled by the feedback loop; maybe they are slightly off by one of two degrees spatially). In this case, the radiation shield would be at a lower temperature than the hot junction heat clamp. Looking at the image in Fig. S1, I suspect it could be that the insulating spacers (Fig. 2a) may not be insulating enough to prevent the shield from being cooled (maintaining a uniform temperature equal to the hot clamp temperature everywhere) or the Peltier cooler is radiatively cooling the radiation shields or hot clamp (which I suspect as the degree of parasitic heat loss correlates with a decrease in length indicative of a view factor change). Since the hot junction temperature is fixed and the cold clamp temperature is lowered, there would then be a systematic over approximation of the heat transferred from the joule heater. Hence, the measured value would increase as the temperature difference increases (thereby artificially increasing the slope) and the thermal conductivity of the sample would be overestimated. In the case of radiatively cooling the radiation shield. This overestimation would be non-linear with the temperature difference over a wide range but linear over the small temperature differences in this study and would systematically scale with the increase in thermal conductivity of the sample.

Response (7): The reviewer's analysis is misguided by the misinterpretation of the y-intercept in Fig. 2b (see response (1)). The sample radiation effect on the measurement is discussed in detail in the SI. The radiation shield is made of copper and all walls are at least 3 mm thick. The temperature of the shield is controlled with an electrical heater. The Biot number is in the 10^{-5} scale for the shield and the presence of a radiation heat flux. Thus, it is extremely unlikely that the cooling effect from the cold-side clamp will affect the shield temperature. With respect to the possible heat transfer between the radiation shield and the liquid-cooled copper stage; that heat transfer is minimized by using 3 double-hole Zirconia (thermal conductivity ~ 1.5 W/m-K) tubes with a diameter of 2 mm. It is important to realize, however, that in this experimental method potential temperature gradients within the radiation shield and temperature reading errors between the shield and heater clamp are not of concern if those temperature differences stay constant during the experiment. The liquid-cooled copper cold stage used as the heat sink spreads the changing rejected/absorbed heat from the Peltier module very efficiently suggesting a constant cold-stage temperature. Our measured sample conductance values are around 0.4 mW/K. Thus, changing the temperature difference along a suspended sample by 1 K result in a change in rejected/absorbed Peltier heat of ~ 2.4 mW (assumed COP = 0.2). Our copper ($k = 380$ W/m-K) cold plate is 1 cm thick, 5 cm wide and the distance between the Peltier module and the

liquid cooled cooling stage (Aluminum plate with copper tubes) is ~20 cm. Thus, for each degree K change along the suspended sample (resulting in a rejected Peltier heat flux of 2.4 mW) the copper cold plate only changes by less than 0.003 K ($\Delta T = (\text{Peltier heat}) \cdot \text{length} / (\text{width} \cdot \text{thickness} \cdot k)$). Consequently, it is reasonable to assume a constant cold-stage temperature, constant heat transfer through the Zirconia tubes and a constant temperature distribution in the radiation shield (Details: copper cold plate (380 W/m-K) is 1 cm thick, 40 cm wide and the distance between the Peltier module and the liquid cooled stage (Aluminum plate with copper tubes) is ~20 cm).

Comment (8):

In the case of (ii), there would exist a bypass heat pathway (potentially radiation) between the hot clamp and cold clamp. As the temperature difference between these clamps increases so would the amount of bypass heat. Hence, the measured value would increase as the temperature difference increases (thereby artificially increasing the slope) and the thermal conductivity of the sample would be overestimated.

Response (8): The effect of thermal shunting on the measurement accuracy is discussed in the SI. We performed calibration experiments without a suspended sample after the sample measurement to quantify the radiative thermal shunting heat flow. As mentioned in the manuscript, this radiative heat flow did not exceed 25% compared to the conducted through sample.

Comment (9):

Thus, in both cases the thermal conductivity of the sample would be overestimated. However, the authors performed a single TDTR measurement and found that the thermal conductivity was ~32 W/m-K. While TDTR is a challenging technique to perform on these samples, why not validate the claim of 62 W/m-K (or essentially anything greater than ~30 W/m-K) with TDTR? This seemed very odd to me and for such a claim as “Nanostructured Polymer Films with Metal-Like Thermal Conductivity”; that validation should be performed on all samples exceeding 30 W/m-K. Furthermore, why not validate this with more standard thin-film techniques like 3-omega rather than the hassle and potential error with microtoming? In-plane and through-plane thermal conductivity can be measured by varying heater line widths using 3-omega (I can point to Rev. Sci. Instr. manuscripts that demonstrate this if the authors are unfamiliar). Or using TDTR or FDTR, in-plane thermal conductivity can be measured using the beam offset technique (I can point to Rev. Sci. Instr. manuscripts that demonstrate this if the authors are unfamiliar). Even transient grading or laser flash could validate this claim. It seems very odd that such claims are made hinging on a homebuilt (non-standard) technique that is utilized outside its measurement range where radiation potentially dominates, when alternative options are available.

Response (9): All of the reviewer's concerns with respect to the accuracy of our method and further data interpretations are based on the misunderstanding of the monotonically increasing y-intercept in Fig. 2b and on a sample length that is 1-2 orders of magnitude larger than the real sample lengths (Fig. S1). We hope that our explanations have already addressed his/her concerns. His/her statement that our technique is outside its measurement range is simply false because we accurately measured the reference samples (Dyneema⁵, Zylon⁵, and stainless steel⁶) with similar thermal conductance values to support our results. We openly and transparently discuss and quantify (theoretically and experimentally) the effects of parasitic heat losses on the measurement in the SI and we are fully aware of the limitations of this method and setup. However, based on our analysis, the calibration experiments, and reference sample measurements (Dyneema⁵, Zylon⁵, and stainless steel⁶), the measured sample conductance values are well within the methods capability.

The corresponding author's group has done extensive work in 3-omega and TDTR and is familiar with all these techniques. We encountered issues such as surface roughness and metal adhesion issues in trying different methods, and believe that the method we used is the most appropriate for our samples.

In summary, I have very serious concerns with the thermal conductivity measurements as I suspect that radiation is the dominant heat loss mechanism (not conductance through the polymer film) and is the source of the systematic error (parasitic heat loss and y-intercept). Thus, I think the reported value of thermal conductivity is overestimated. This then brings into question the subsequent analysis and conclusions.

To improve the manuscript, to refute these observations, and to convince this reviewer of the validity of the measurements and conclusions of this work, please perform all of the following: 1) Share the data of the control experiments where no sample is present where the clamp heater is held at various heights above the cold clamp to demonstrate radiation is not significant.

We added figures (Fig. S1c-d) in the SI to show the directly measured heating electrical power P_{el} (raw data, with the expected random y-intercepts). We have also included the calibration data to show the radiative thermal shunting heat flow between the heater and cold-side clamp. The measured radiative thermal shunting between the heater and cold-side clamp does not exceed 25% of the heat conducted through the sample and we always correct for that heat flow.

2) Perform a thermal conductivity measurement of a copper foil using the homebuilt steady-state system as a function of length, reporting the thickness and width (including a photograph).
3) Perform measurements on standards of (opaque) films ranging from 30 W/m-K to 60 W/m-K with the homebuilt technique (without systematic error).

We measured several reference samples (Dyneema⁵, Zylon⁵ and stainless steel⁶) of similar geometry and thermal conductance values as our drawn polymer samples (Fig. R2c, the same as Fig. S1e). Copper foil itself has much higher thermal conductivity and low emittance, and therefore is not a good representation of our measurement.

4) Explain the systematic error and increase of the large (~2 mW of 10 mW) parasitic heat loss that follows the sample order of increasing draw ratio (or decreasing length, or increasing transparency).

This is based on the misinterpretation of Fig 2b by the reviewer. We added the directly measured heating electrical power P_{el} in the SI showing random y-intercepts (Fig S1c-d).

5) Perform TDTR (beam offset), 3-omega (multiple heater line width), transient grating, or laser flash measurements on the samples with draw ratios of 30, 40, 60, 70, 80, 90, 100, and 110. Try for anisotropic measurements using beam offset or multiple heater line widths.

We are glad to see that the reviewer shares our emphasis on always presenting accurate data. However, we respectfully disagree with him/her that further experiment is needed to support our conclusions. A list of reasons is provided below.

1) The reviewer's suspicions of systematic errors in our measurements are ultimately based on his/her misunderstanding of Fig. 2b and Fig. S1. As explained above and noted in the original caption of Fig. 2, Fig. 2b plots geometrically scaled raw data which, when taken directly as the raw data (Fig. S1c-d), leads to various evaluations of the technique that are incorrect. Further, we emphasize that the images in the original Fig. S1 are presented to better illustrate our setup, instead of showing the actual dimensions. We have, however, shown the details of the sample geometry in Table S1. We have also added an image in Fig. S1 to better represent our setup for measuring the polyethylene films (70 ×).

2) The accuracy of our measurements is first of all demonstrated by our measurements of several carefully selected reference samples (Dyneema⁵, Zylon⁵ and stainless steel⁶) which agree well with literature values. Such accuracy is achieved via careful considerations and management of radiation and conduction heat losses, as well as radiation shunting between the hot and cold clamps.

3) Finally, we note that we made great effort to laminate the films and measure with TDTR, not simply to check the validity of our steady-state measurement. More importantly, our goal is to show that these films can be laminated to achieve much larger thickness while maintaining their high thermal conductivity. Such robustness is valuable towards its industrial applications. On the other hand, we also note that techniques like TDTR, 3-omega, transient grating, and laser flash

are yet to be recognized as the standards for polymer thermal transport measurement. All have their own merits but caveats as well. We are familiar with these techniques and have them in our own lab, but they did not work well for our samples due to surface roughness, metal adhesion, etc.

6) Redo amorphous thermal conductivity calculations based on this data (Figs. 3b and S15).

We hope by now we have addressed the reviewer's concern and this point becomes moot.

Reviewer #3 (Remarks to the Author):

This work provides important data and understandings regarding thermal conductivity in drawn polyethylene films. The authors find that the films consist of nanofibers with alternating crystalline and amorphous regions, which is consistent with lamellar structure expected for drawn polyethylene. The thermal conductivities of these two regions are not measured directly. The amorphous region's thermal conductivity is derived from fitting experimental data to a 1D thermal model in which the crystalline region is assigned an assumed value based on a first-principles calculation in another work. The authors find that the derived thermal conductivity of the amorphous phase increases with draw ratio, and their overall finding is that thermal conductivity of the amorphous phase is the bottleneck to film thermal conductivity.

In general, I view the topic area of heat conduction in polymers to be an important one, both scientifically and technologically. Furthermore, the structural and thermal data provided by this work will be useful to other researchers studying thermal conductivity in semicrystalline polymers. However, for this particular work I consider general finding that the amorphous areas surrounding lamellar crystals are the primary bottleneck to thermal conductivity to be a reasonably expected result and not of sufficiently broad interest.

Response: We thank the reviewer for acknowledging the significance of our achievement and understanding of high thermal conductivity in PE films. We agree that one would reasonably expect the amorphous phase to be the bottleneck to film thermal conductivity. However, we respectfully clarify that our key contribution in terms of fundamental understanding lies in our revealing of a surprisingly high thermal conductivity in the amorphous phase. As pointed out by Reviewer 1, we came to this conclusion only because we took great effort and managed to “quantitatively using beam-line scattering/structural characterization to understand the evolution of transport processes is new”. Such an approach “was badly needed for this class of materials”, and “will have a high degree of influence on future thinking for these materials and other similar classes of materials”. We hope our revisions and responses could help clarify our contributions.

Reference:

1. Broido, D. A., Malorny, M., Birner, G., Mingo, N. & Stewart, D. A. Intrinsic lattice thermal conductivity of semiconductors from first principles. *Appl. Phys. Lett.* **91**, 231922 (2007).
2. Esfarjani, K., Chen, G. & Stokes, H. T. Heat transport in silicon from first-principles calculations. *Phys. Rev. B* **84**, 85204 (2011).
3. Li, W., Carrete, J., A. Katcho, N. & Mingo, N. ShengBTE: A solver of the Boltzmann transport equation for phonons. *Comput. Phys. Commun.* **185**, 1747–1758 (2014).
4. Henry, A. & Chen, G. High thermal conductivity of single polyethylene chains using molecular dynamics simulations. *Phys. Rev. Lett.* **101**, 235502 (2008).
5. Wang, X., Ho, V., Segalman, R. A. & Cahill, D. G. Thermal conductivity of high-modulus polymer fibers. *Macromolecules* **46**, 4937–4943 (2013).
6. Sweet, J. N., Roth, E. P. & Moss, M. Thermal conductivity of Inconel 718 and 304 stainless steel. *Int. J. Thermophys.* **8**, 593–606 (1987).

Reviewers' comments:

Reviewer #1 (Remarks to the Author):

With regard to the authors responses to my comments:

I understand the capabilities of first-principles calculations as a means to calculate phonon lifetimes and thermal conductivity and I am aware of the authors history in that field (and the MD work). However, this is quite a unusual application of DFT; the validation that has occurred so far in the first-principles calculations is mostly from covalent or ionic bonded crystals. As the cited paper explicitly says (I see that is now published in "Nanoscale"), it is really the first first-principles calculation of the thermal conductivity of a polymer crystal. While I am not an expert in the area, I would say it's a quite unusual system in that the primary interactions dominating interchain bonding are van der Waals bonds...something DFT is not intrinsically designed to handle. Yes, VASP provides pseudopotentials to do this, but I don't think that calculations of third-order IFC's or the derived thermal conductivities calculated this way have been validated by experiments previously.

However, as the author's have argued in their rebuttal, the exact value of the crystalline thermal conductivity actually does not matter (except for the last few points of fig 3b, which is what I had argued in my original comment). I have read the authors comments carefully, and I believe they are correct so long as their reported crystalline fractions, k_{tot} measurements are correct. I see that the 2nd reviewer is carefully scrutinizing the latter point, so let me ask an outstanding question I have about the former: how exactly was the crystalline fraction calculated? I tried reading the SI, but I see little to no discussion about this and it doesn't appear in the paper either. In particular, I surmise that you are extracting this from WAXS data somehow, but when I look at the WAXS patterns provided (a) it's unclear exactly how this would be done with high degree of accuracy; are referencing say the amorphous peak intensity to some reference of known crystallinity? a ratio of crystalline to amorphous? I didn't get it, and it is quite important. (b) for the high draw ratios in particular (say the 110X whose picture is given in 4b), I can see no amorphous ring at low q , yet you were able to assign a finite crystalline fraction (appears to be about 0.92 in 4d). Again, how are you able to extract something-from-nothing? What are the estimated uncertainties (I see now that no errorbars are provided in figure 4d, and this must change before publication). Please address this point.

With regard to Reviewer 2's comments:

I would like to make a few comments about the responses to Reviewer #2, whose comments/author rebuttal I read/followed carefully. On first review I did not notice the systematic offset pointed out by that reviewer, but I do see it now. The authors have written a rebuttal to that comment (apparently, in short, they were rescaling things in a non-transparent way; I will let the 2nd Reviewer sort out whether that response is adequate), but there was something else that caught my eye in terms of suggested corrective actions: the 2nd reviewer suggested that techniques such as TDTR w beam offset or three omega should be used to confirm the homebuilt measurements. These techniques are very challenging for these samples, and would probably not be better than a correct implementation of the current method. The 3-omega technique requires the ability to microfabricate a heater (and a rather thin heater if you want sensitivity to anisotropy; the assumption of constant heat flux is also questionable for a high k metal film on a low k substrate) and make electrical connections to it, rather hard when the metal film adhesion is low and the sample geometry is unconventional; this is quite challenging for polymeric/non-standard samples and may even alter the sample. Similarly the TDTR offset technique might be difficult on this sample because metal transducers themselves have a large in-plane conductance and the vertical thermal penetration depth of heat would necessarily be small for PE because of the low thermal conductivity in other directions. I have not examined the sensitivity analysis in this situation, but I do not think it would be favorable to beam offset TDTR. I definitely think that we should scrutinize the current thermal measurement method, but I do not think that some

alternative method of experiments would be a productive or necessary venture.

Reviewer #2 (Remarks to the Author):

I'm grateful to the authors for the deep consideration of my previous reviewer comments. I am particularly grateful for the addition of the deeper radiation analysis as I felt details were lacking to rule out those effects that has now been provided. The authors are correct; it was unclear to me that the y-axis in Fig 2b was the geometrically scaled heating power as it was mislabeled in the original figure. Thanks to the authors for correcting the figure axis and adding to the caption. However, I must now insist that Fig. 2b be replaced entirely with a modified and combined version of Figs. R2a-b. The primary reason for this is - and I use the author's own exact words in their Response (6) - "the y-intercept in Fig. 2b is meaningless due to the geometric scaling." I agree with the authors, this arbitrarily scaling of data is meaningless (why not scale it with 30x or 100x?); the raw unaltered data should be presented with extrapolation to zero ΔT .

The primary claim of this manuscript is based on this data. Frankly, my expectation for publication in a quality journal is that meaningful, truthful (e.g., unscaled, unaltered, transparent) data be presented in the primary manuscript.

I find it curious that 50x was chosen for the "meaningless" geometric scaling as this is the only draw ratio where the thermal conductivity is verified with TDTR and for which is the point the thermal conductivity begins to exceed expectations (e.g., the main claim of the manuscript). I find this peculiar and question if the main finding of the manuscript is merely a (mis-) interpretation caused by this scaling.

While I'm appreciative of the authors thorough and detailed response to reviewer comments in general, I expected that much more of the discussion in the review process would have made it into the updated main manuscript (or SI), as it identifies weaknesses in the original manuscript. For example, I greatly appreciate the reporting of the sample conductance and I'd appreciate inclusion of the logical discussion on radiation and parasitic heat losses. This is missing in the manuscript and the manuscript is currently written in a way that expresses "just trust me" rather than "here is the evidence." Please strongly consider major modifications to the manuscript to include our discussion for the benefit of future readers.

Finally, I am still not convinced that measurements were performed properly and that the reported data/findings are not erroneous. I greatly appreciate the additional statements around calibration samples that was missing in the original manuscript and have now been included. However, my six numbered requests were largely ignored or dismissed and data has not been presented as requested. I reiterate these requests below:

- 1) My numeric comment 1 was dismissed. To be explicit about my previous request, please provide measured electrical heating power as a function of temperature difference for varying hot and cold clamp distances, L . This should take very little time to perform and is of negligible cost.
- 2) My numeric comment 2 was dismissed. I want to see the measured electrical heating power versus temperature difference of a highly conductive film (e.g., copper foil) as a function of length (reporting the thickness and width, and a photograph). This should take very little time to perform and is of negligible cost.
- 3) My numeric comment 3 was dismissed. Validate measurements with (opaque) films in the range of 30 W/m-K to 60 W/m-K. The authors state that validation was performed on stainless steel 304 (~15 W/m-K) and in Fig 2Ra it clearly has the steepest slope (because the conductance is ~2.1 W/K where the polymers in this study have conductance's around 0.4 W/K). Validation was also performed on Zylon and Dyneema but with conductance is around 1.3 W/K. I'd strongly suggest that when the requested validation measurements are performed, that the geometry is appropriately adjusted to maintain the same approximate overall conductance (0.4 W/K), as was conducted with the drawn films. (Side note: rather than adjusting the geometry, a more

convincing study would be if the geometry was fixed (e.g., cut the samples to the same smallest size), as it would more clearly show how only the thermal conductivity was changing, not both geometry and conductivity. This would have been a better plan of experiments from the beginning.)

5) My numeric comment 5 was dismissed. I must adamantly insist that these other measurements be performed: TDTR (with beam offset), 3-omega (with multi-heater lines width), transient grating, and/or laser flash (with film in-plane measurements). I am glad that the authors "are familiar with these techniques and have them in our [their] own lab..." I therefore expect that they should be able to perform the requested measurement quickly and with little cost burden. I am also intimately familiar with the merits and caveats of these techniques, which is why I explicitly suggested this subset of techniques. The authors have already performed TDTR experiments and report on the results for 50x in this manuscript (despite adhesion challenges), so that should be doable to perform on the higher conductivity samples (e.g., 100x). 3-omega can be insensitive to surface roughness and I find it hard to believe that the films are too rough (these are drawn polymer films after all). Transient grating may be challenging given opacity or adhesion, but I leave it as an option. In-plane laser flash is trivially easy and extremely well suited for this measurements and sample geometry given the claimed high thermal conductivity (and adherence to an ASTM standard). The commercial vendor Netzsch has a laboratory in proximity to the corresponding author's lab and they would happily perform these measurements at no cost; I'm even willing to call Netzsch on the author's behalf if necessary. Authors can use Netzsch thin-film holder, coat the sample with graphite paint, and even correct for transparent films effects. Measurements can be completed within hours at negligible cost.

6) My numeric comment 6 was dismissed. Based on the data to be provided the amorphous thermal conductivity calculations may still need to be redone.

I will not yield my position that this manuscript is not appropriate for publication until I see the requested data. These researchers are uniquely situated to perform these additional measurements since they are the only group with access to these drawn films. This characterization should be performed for the betterment of the entire thermal science community, for which the corresponding author is an admired leader and trusted pillar. Very frankly, I would like to avoid another spider-silk or polyethylene-nanofiber high thermal conductivity measurement fiasco. Research resources are finite and this extra effort up-front could save others significant time not trying to reproduce erroneous results of high thermal conductivity.

Reviewer #3 (Remarks to the Author):

First I'd like to thank authors for clarifying their contribution, which is the "revealing of a surprisingly high thermal conductivity in the amorphous phase." I have a few further questions to verify this point.

As the draw ratio increases, "amorphous" region develops orientation order. This is expected, as it is becoming more crystalline. Given this increasing order, is it still valid to term this region "amorphous"?

On a related note, do the X-ray scattering measurements have sufficient signal-to-noise to be able to compare level of structural disorder in these regions to the structural disorder in bulk polymers commonly referred to as "amorphous"?

The manuscript states that k_a is high compared to "typical 0.3 W/m-K", but wouldn't that 0.3 value be associated with films that have no orientation? If the orientational order here is increasing, why is region's increase in thermal conductivity surprising?

The authors use a periodic 1D thermal model, including in their sensitivity analysis. Could the authors clarify how they know that the regions are always only in series and never in parallel? Figure 1d shows the regions as being perfect columns, or discs in cross-section. How sure are the authors that this is valid? This is important to verify given extremely large difference in thermal conductivity assumed for the two regions. If the crystalline regions were to penetrate the amorphous regions at all (for example, small pinholes that acted as thermal shunts), or if they were to tilt at an angle, so that a given cross-section of the fiber would have both crystalline and amorphous regions, the thermal model would appear to be invalid.

Finally, I would also like to ask authors to consider revising their abstract and concluding paragraph ("In summary...") to better reflect their stated contribution. As it currently reads, it appears they are saying that their contribution is the knowledge that the amorphous phase dictates overall thermal conductivity, which as they have agreed in their response is what one would normally expect.

Revision Report for NCOMMS-17-22345A

Nanostructured Polymer Films with Metal-like Thermal Conductivity

We thank the reviewers for their careful reading and constructive criticism. Reviewers' inputs are very helpful for improving our manuscript. Our point-by-point responses are documented below. The reviewers' original comments are in black and our responses are in blue.

Reviewer #1

With regard to the authors responses to my comments:

I understand the capabilities of first-principles calculations as a means to calculate phonon lifetimes and thermal conductivity and I am aware of the authors history in that field (and the MD work). However, this is quite a unusual application of DFT; the validation that has occurred so far in the first-principles calculations is mostly from covalent or ionic bonded crystals. As the cited paper explicitly says (I see that is now published in "Nanoscale"), it is really the first first-principles calculation of the thermal conductivity of a polymer crystal. While I am not an expert in the area, I would say it's a quite unusual system in that the primary interactions dominating interchain bonding are van der Waals bonds...something DFT is not intrinsically designed to handle. Yes, VASP provides pseudopotentials to do this, but I don't think that calculations of third-order IFC's or the derived thermal conductivities calculated this way have been validated by experiments previously.

We agree with the reviewer that as thermal conductivity of polymer crystals is yet to be measured, the current computational technique has not been directly verified in polymer systems. Also, we want to mention that if there is any effect mimicking the Fermi-Pasta-Ulam mechanism, the current approach based on DFT and Boltzmann transport equation will also underestimate the thermal conductivity. On the other hand, we also note that the effect of van der Waals interaction on

thermal transport has been tested in other layered materials, such as graphite's anisotropic thermal conductivities, and that the calculation is the best that can be done by the scientific community at this stage. Applying the DFT results in our analysis represents a decent effort to extract the transport physics. We want to emphasize that, as the reviewer agrees (see next comment) the extracted k_a is not sensitive to exact values of the crystalline phase thermal conductivity as long as the latter is high enough.

Revisions made: we have added caution in the manuscript (page 9) on the status of DFT.

We do caution that this estimation is subject to uncertainties in k_c as first principles simulation combined with Boltzmann transport equation cannot capture any non-ergodic behavior that may exist in such systems, and is also not well suited for materials with van der Waals interactions. However, the estimated k_a will not be sensitive to the exact value of crystalline thermal conductivity (see discussion in SI 1.4.2 and Fig. S16).

However, as the author's have argued in their rebuttal, the exact value of the crystalline thermal conductivity actually does not matter (except for the last few points of fig 3b, which is what I had argued in my original comment). I have read the authors comments carefully, and I believe they are correct so long as their reported crystalline fractions, k_{tot} measurements are correct. I see that the 2nd reviewer is carefully scrutinizing the latter point, so let me ask an outstanding question I have about the former: how exactly was the crystalline fraction calculated? I tried reading the SI, but I see little to no discussion about this and it doesn't appear in the paper either. In particular, I surmise that you are extracting this from WAXS data somehow, but when I look at the WAXS patterns provided (a) it's unclear exactly how this would be done with high degree of accuracy; are referencing say the amorphous peak intensity to some reference of known crystallinity? a ratio of crystalline to amorphous? I didn't get it, and it is quite important. (b) for the high draw ratios in particular (say the 110X whose picture is given in 4b), I can see no amorphous ring at low q , yet you were able to assign a finite crystalline fraction (appears to be about 0.92 in 4d). Again, how are you able to extract something-from-nothing? What are the estimated uncertainties (I see now that no errorbars are provided in figure 4d, and this must change before publication). Please address this point.

We discussed how to extract the crystallinity in SI (section 1.3.2 in the previous submission and same section 1.3.2 in the revision here). We used WAXS data to extract the crystallinity. This section has now been revised and we provide below more explanation.

The determination of fraction crystallinity of polymers is often done with measurements of specific volume, specific heat, specific enthalpy, infrared extinction coefficients, linewidths of NMR (nuclear magnetic resonance), XRD (X-ray diffraction) intensities etc¹. There is some inconsistency when comparing the results of different methods because each method relies on a specific property of the materials. XRD is one of the mostly widely used methods and it is based

on the assumption that the number of elastically scattered photons by one phase is proportional to the amount of that phase in the scattering volume. This leads to another requirement for this method to work, i.e. tagging each scattered photon to the corresponding phase (crystalline or amorphous)². The tagging of photons can be readily achieved provided a known crystalline lattice type and structure². As given in the equation 1 below (same as equation 8 in SI) following standard polymer science textbooks² (for example, reference 41 in SI: Kasai, N. & Masao, K. X-ray diffraction by macromolecules, Springer, 2005),

$$\text{Crystallinity} = \frac{I_{\text{crystalline}}}{I_{\text{crystalline}} + I_{\text{amorphous}}} \quad (\text{equation 1})$$

Briefly, the $I_{\text{crystalline}}$ and $I_{\text{amorphous}}$ are obtained by fitting the total intensity of the each 1D XRD pattern (after up-mapping to a full 2D pattern, necessary pixel-by-pixel based geometric and instrumentation corrections to the raw intensities, and reduction to 1D line profile; see reference³ same as reference 40 in the SI) to the incoherent sum of the integrated intensity (area under the curve) of Bragg peaks for the crystallite contribution and broad amorphous peaks for the amorphous contribution. During the fitting, each peak is usually modeled by a Voigt function (a convolution of Gaussian and Lorentzian). After the quantitative separation of the crystalline and amorphous peaks (Fig. S9b), the crystallinity is obtained via the equation above using the integrated area calculated via the close-form Voigt formulas with parameters obtained from fitting.

Given known crystalline lattice types and expected Bragg and amorphous peak positions, the crystalline and amorphous contributions can be very well separated. Confident crystallinity using this XRD method can be obtained given sufficient photon statistics. We caution that the definition of crystallinity varies across different techniques as mentioned above, so one must be aware that the error bar or uncertainty analysis is biased towards the adopted technique and does not provide useful information about the deviation from the “true” crystallinity. Instead, this error bar analysis only indicates the confidence of the given measurement and data analysis conditions as compared to the best capability provided by the scope of technique itself (for example, XRD with sufficient photons, perfect diffraction angular resolvability, etc.). In terms of the crystallinity interpretation, as long as the scope of measurement method is confined with a single technique, the comparison of different crystallinities is often very reliable provided sufficient photon statistics for XRD method.

On large draw-ratio samples, the diffraction intensities from both crystalline and amorphous intensities gradually decreases from bulk samples due to thinner samples (less amount of materials) in the beam, therefore requiring a very intense X-ray source that is far brighter than X-ray instrumentations in university laboratories. This is the reason why the synchrotron radiation facility is used in this work. With lab X-ray sources the XRD intensity is typically weak and noisy from polymeric samples, therefore, thus often requiring error bars analysis for the uncertainty of the integrated area under the Bragg peaks. However, this is not often necessary for many

synchrotron data analyses unless insufficient photon statistics is of a concern (which is not the case here, seen as follows).

On high draw-ratio samples, for example, as in the 110 \times sample, while unprocessed 2D pattern does not reveal significant amorphous contribution, sufficient photon statistics can be seen after the data is converted to 1D XRD curve (following Figs. R1a and R1b) and thus a high confident estimation of the crystallinity is warranted. We added more discussion on crystallinity calculation in SI 1.3.2 WAXS analysis.

a

b

Figure R1. The crystallinity analysis for the drawn polyethylene at 110 \times draw ratio. (a) linear scale. (b) log scale to enhance the significance of amorphous contribution to the crystallinity determination.

While XRD gives reasonable crystallinity estimation, it does not provide information about the sizes and distributions of crystalline and amorphous regions, which is important for our thermal modeling. We use small-angle X-ray scattering analysis (Fig. 4, Fig. S10 and Fig. S11) for such purpose. Such analysis gives a *statistical* description of the crystalline and amorphous size and positional distributions via modeling of the form factors and structure factors of the oriented lamellas (See manuscript and SI 1.3.3 SAXS analysis, Fig. S10 and Fig. S11).

With regard to Reviewer 2's comments: I would like to make a few comments about the responses to Reviewer #2, whose comments/author rebuttal I read/followed carefully. On first review I did not notice the systematic offset pointed out by that reviewer, but I do see it now. The authors have written a rebuttal to that comment (apparently, in short, they were rescaling things in a non-transparent way; I will let the 2nd Reviewer sort out whether that response is adequate), but there was something else that caught my eye in terms of suggested corrective actions: the 2nd reviewer suggested that techniques such as TDTR w beam offset or three omega should be used to confirm the homebuilt measurements. These techniques are very challenging for these samples, and would probably not be better than a correct implementation of the current method. The 3-omega technique requires the ability to microfabricate a heater (and a rather thin heater if you want sensitivity to anisotropy; the assumption of constant heat flux is also questionable for a high k metal film on a low k substrate) and make electrical connections to it, rather hard when the metal film adhesion is low and the sample geometry is unconventional; this is quite challenging for polymeric/non-standard samples and may even alter the sample. Similarly the TDTR offset technique might be difficult on this sample because metal transducers themselves have a large in-plane conductance and the vertical thermal penetration depth of heat would necessarily be small for PE because of the low thermal conductivity in other directions. I have not examined the sensitivity analysis in this situation, but I do not think it would be favorable to beam offset TDTR. I definitely think that we should scrutinize the current thermal measurement method, but I do not think that some alternative method of experiments would be a productive or necessary venture.

We thank the reviewer 1 for his/her objective and reasonable assessment of different methods “the 2nd reviewer suggested that techniques such as TDTR w beam offset or three omega should be used to confirm the homebuilt measurements. These techniques are very challenging for these samples, and would probably not be better than a correct implementation of the current method” and “I definitely think that we should scrutinize the current thermal measurement method, but I do not think that some alternative method of experiments would be a productive or necessary venture”.

In the reply to Reviewer 2 part, we have provided carefully-performed additional measurements to justify the ability of our home-built system in measuring the thermal conductivity of our drawn polyethylene films. We hope these measurements have clarified the questions regarding our set up.

Reviewer #2:

I'm grateful to the authors for the deep consideration of my previous reviewer comments. I am particularly grateful for the addition of the deeper radiation analysis as I felt details were lacking to rule out those effects that has now been provided. The authors are correct; it was unclear to me that the y-axis in Fig 2b was the geometrically scaled heating power as it was mislabeled in the original figure. Thanks to the authors for correcting the figure axis and adding to the caption. However, I must now insist that Fig. 2b be replaced entirely with a modified and combined version of Figs. R2a-b. The primary reason for this is - and I use the author's own exact words in their Response (6) - "the y-intercept in Fig. 2b is meaningless due to the geometric scaling." I agree with the authors, this arbitrarily scaling of data is meaningless (why not scale it with 30x or 100x?); the raw unaltered data should be presented with extrapolation to zero ΔT . The primary claim of this manuscript is based on this data. Frankly, my expectation for publication in a quality journal is that meaningful, truthful (e.g., unscaled, unaltered, transparent) data be presented in the primary manuscript.

Although we do not necessarily agree with the reviewer's comment as we believe the paper should be written clearly and figure should be easy to understand, we reluctantly replace now the raw data (following Fig. R1a) in the main text Fig. 2b. The geometric scaling (following Fig. R1b) is removed from the primary manuscript, and it is in supplementary information Fig. S1. All the characterizations related with electrical heating power — temperature are transparent in the whole manuscript and supplementary information.

Figure R1. (a) Directly measured electrical heating power (P_{el}) for all the polyethylene films and representative calibrations. (b) Geometrically scaled electrical heating power (P_{els}) as a function of the temperature difference ($T_h - T_c$) across films. Representative data (10 \times , 90 \times , 110 \times , Dyneema and S. Steel 304) are scaled to the geometry of a 50 \times film ($P_{els} = P_{el} \cdot (A/L)_{50\times} / (A/L)$). A larger slope indicates a higher thermal conductivity.

I find it curious that 50x was chosen for the “meaningless” geometric scaling as this is the only draw ratio where the thermal conductivity is verified with TDTR and for which is the point the thermal conductivity begins to exceed expectations (e.g., the main claim of the manuscript). I find this peculiar and question if the main finding of the manuscript is merely a (mis-) interpretation caused by this scaling.

We regret that the reviewer does not understand our explanation and suspect that we are “(mis-) interpreting” the data, despite the effort we put into the previous response. We would like to make it clear that there is no data misinterpretation as the reviewer casually suggested. We will discuss more on reliability and validity evidences of thermal conductivity measurements in the following sections.

1. The fact that we choose 50 \times as the “meaningless” geometric scaling has nothing to do with the TDTR results.
 - i. For the 50 \times geometric scaling: we choose a sample with 50 \times draw ratio that is in the middle range of thermal conductivity and draw ratio, to show the representative sample/data.

- The geometric scaling was intended as a more intuitive presentation of the raw experimental data. Detailed sample dimensions were provided from the very beginning as Table S1 (same as Table R1a now) in the supplementary information so the information was complete.
 - All the thermal conductivity values reported via the steady-state method and TDTR were independently measured and have nothing to do with the scaling process.
- ii. For the concern of the reviewer on the TDTR measurements, we make the following remarks.
- In principle, it is not necessary for us to do TDTR measurement, because our steady-state system is well validated by extensive reference sample tests and careful error analysis (Fig. S1 and Fig. S2, SI 1.2 thermal conductivity measurements, and following discussion), we are confident that our measured results are reliable.
 - In reality, we dedicatedly double check steady-state results, and it is reasonable for us to use 50× drawn polyethylene films.
 - As already mentioned in supplementary information, 1.2.2 time-domain thermoreflectance method section, it is very challenging to make a polymeric sample with optically smooth surface — root-mean-square roughness of the polyethylene surface to be ~10 nm in a $15\ \mu\text{m} \times 15\ \mu\text{m}$ region (Fig. 2c and Fig. S6), we choose the relatively easy 50× sample with high thermal conductivity.
 - In order to measure the thermal conductivity along the chain direction by TDTR, we first need to laminate 100-layer aligned polyethylene films by hot pressed method (Fig. 2c), embed laminated polyethylene films into epoxy matrix, then microtome the cross-section surface (Fig. S6a-c). We make sure the film roughness less than 10 nm (Fig. S6d). We spend six months to prepare only one successful ultra-smooth sample by microtoming. It is quite challenging to achieve such ultra-smooth and aligned polymeric films without altering the sample.
 - The difficulty in sample preparation for TDTR is further complicated by the fact that the yield of our drawing platform is not very uniform at higher draw ratio (the higher draw ratio we stretch, the more un-uniform film we have). The typical thickness of single 50× and 110× film is ~1.5 μm and ~0.6 μm , respectively. To have the ~150 μm thick sample, we choose relatively thick sample (50×) for lamination.

Table R1. Measured sample dimensions.

(a) Measured film dimensions at various draw ratios.

(b) Measured standard samples, including 304-stainless steel foils⁴, Zylon fibres⁵, Dyneema fibres⁵, Sn⁶ and Al⁶ films.

(a)

Draw ratio	Length (mm)	Cross-sectional area (mm ²)	Thermal conductance (W/K)
1	1.3	0.6876	2.19E-04
2.5	1	0.10395	5.45E-04
5	1.1	0.06435	3.73E-04
10	2	0.055858	3.34E-04
20	1.5	0.050818	5.08E-04
30	2	0.028584	3.85E-04
40	2.2	0.043407	4.65E-04
50	2.25	0.021630	3.21E-04
60	2.2	0.016458	2.53E-04
70	1.65	0.0083974	2.32E-04
80	1.15	0.010078	4.68E-04
90	1.1	0.0075624	3.81E-04
100	0.65	0.0068800	6.02E-04
110	1	0.0052500	3.58E-04

(b)

Standard sample	Length (mm)	Cross-sectional area (mm ²)	Thermal conductance (W/K)
Dyneema	1.36	0.06810	1.21E-03
Zylon	1.21	0.07000	1.31E-03
S. Steel 304	3.2	0.4725	2.24 E-03
Sn	5.55	0.03425	4.48 E-04
Sn	3.80	0.02500	5.08E-04

Sn	2.55	0.01300	3.85E-04
Al	5.79	0.01285	4.92E-04
Al	4.30	0.01285	6.57E-04

While I'm appreciative of the authors thorough and detailed response to reviewer comments in general, I expected that much more of the discussion in the review process would have made it into the updated main manuscript (or SI), as it identifies weaknesses in the original manuscript. For example, I greatly appreciate the reporting of the sample conductance and I'd appreciate inclusion of the logical discussion on radiation and parasitic heat losses. This is missing in the manuscript and the manuscript is currently written in a way that expresses "just trust me" rather than "here is the evidence." Please strongly consider major modifications to the manuscript to include our discussion for the benefit of future readers.

We updated our manuscript, added more discussions on radiation and parasitic heat losses in both the main text and methods section. We further added discussion on radiation error analysis and parasitic heat losses in supplementary information (section 1.2 thermal conductivity measurements) and supplementary figures (Fig. S1 and Fig. S2). We attached marked manuscript and supporting information to show where revisions were made. We remind the reviewer that the paper has length limit and believe additional experimental details in the SI are still valuable.

Finally, I am still not convinced that measurements were performed properly and that the reported data/findings are not erroneous. I greatly appreciate the additional statements around calibration samples that was missing in the original manuscript and have now been included. However, my six numbered requests were largely ignored or dismissed and data has not been presented as requested. I reiterate these requests below:

1) My numeric comment 1 was dismissed. To be explicit about my previous request, please provide measured electrical heating power as a function of temperature difference for varying hot and cold clamp distances, L . This should take very little time to perform and is of negligible cost.

According to reviewer's suggestions, we carefully designed and properly perform measurements on both calibration tests (without mounting any samples) and reference Sn⁶ and Al⁶ samples with high thermal conductivity (following Fig. R2a; Fig. R3a, same as Fig. S1a). Table S1 (same as Table R1) shows the detailed dimensions for these samples. We measured electrical heating power as a function of temperature difference by varying distances (L) between hot clamp and cold clamp. Fig. R2a (same as Fig S1c) shows experimental results— electrical heating power for reference samples and calibration tests. Fig. R2b (same as Fig. S1d) shows the measured conductance ~ 0.4 mW/K for drawn polyethylene films. Fig. R2c (same as Fig. S1e) shows the measured conductance ~ 0.5 mW/K for reference Sn⁶ and Al⁶ samples. And Fig. R2d (same as Fig. S1f) shows the measured conductance ~ 0.05 mW/K for calibration tests. The conductance value across the gap in

the calibrations tests decreases as the gap distance increases because of the smaller view factor and thus less radiative thermal shunting. However, we note that the measured conductance for drawn polyethylene films and Sn and Al films is ~ 10 times higher than that of this calibration tests. In general, these results show that the conductance errors are less than 20% in our measurements and is less than 10% for most samples.

Regarding with reviewer’s “this should take very little time to perform and is of negligible cost”, we respectfully disagree with his/her comments. It is known that achieving steady-state conditions for steady-state measurement is a time-consuming procedure. And steady-state measurement is a relatively expensive procedure, as it requires more elaborate experiment set up^{7,8}. The fact that this revision took over six months (from February 14, 2018 to August 25, 2018) partially results from this fact (some people left group and training new people is another reason, to be clear).

a

b

c

Figure R2. **a** Directly measured electrical heating power (P_{el}) for all reference samples and calibration tests without mounting samples. **b** Measured thermal conductance of all the polyethylene films. **c** Measured thermal conductance of all the reference samples: Dyneema fiber⁵, Zylon fiber⁵, stainless steel film⁴, Sn film⁶ and Al⁶ film. **d** Measured thermal conductance of calibration tests. The conductance value across the gap in the calibrations tests decreases as the gap distance increases because of the smaller view factor and thus less radiative thermal shunting. Detailed analysis of geometric uncertainties is discussed in 1.2 section: thermal conductivity measurements in the supplementary method.

2) My numeric comment 2 was dismissed. I want to see the measured electrical heating power versus temperature difference of a highly conductive film (e.g., copper foil) as a function of length (reporting the thickness and width, and a photograph). This should take very little time to perform and is of negligible cost.

We want to make it clear, according to Reviewer 2' comments, we could not measure a highly conductive copper foil with thermal conductance ~ 0.5 mW/K using our systems, because it is extremely hard for us to handle and mount such thin and narrow copper foil (5-20 microns thickness and ~ 0.1 mm width, sample from Sigma-Aldrich). Al foil is a good reference sample for our test as the literature thermal conductivity for Al is 237 W/m-K⁶. But Al foil is already very hard to handle (50 microns thickness and ~ 0.5 mm width). Manually mounting Al sample often takes many tries (~ 10 times at least, Fig. R2a, same as Fig. S1a) to be successful.

We additionally measured electrical heating power versus temperature difference of a highly conductive Al⁶ films as a function of length (Fig. R2a and Fig. R3a, which are same as Fig S1c and Fig. S1a). Fig. R3a shows sample dimensions and setup for measuring thermal conductivity.

Table S1 (same as Table R1) shows the detailed sample dimension including length, width and thickness. We kept the thermal conductance for Al films (Fig. R2c, same as Fig. S1e) similar to those for drawn polyethylene films (Fig. R2b, same as Fig. S1d). The measured thermal conductivity Al films are shown in Fig. R3b (same as Fig. 1h): $\sim 202.67 \text{ W/m-K}$ (Al length 4.3 mm) and $\sim 201.89 \text{ W/m-K}$ (Al length 5.79 mm). We note that our measured thermal conductivity is relatively lower than the literature thermal conductivity 237 W/m-K ⁶, which is due to the thermal interface resistance between the measured sample and the copper clamps during the measurements (Fig. R3a, same as Fig. S1a). We minimize thermal interface resistance by using thermal paste (Fig. S3a and S1a), and we discussed thermal contact resistance effects in supplementary information, 1.2.1 Steady-state method minimization of thermal interface resistance. Because there is an effect of thermal contact resistance, we thus UNDERESTIMATED thermal conductivity of all the samples including drawn polyethylene films.

a

b

Figure R3. a Experimental setup photos showing reference samples suspended between the hot and cold copper clamps. Reference samples (Al films⁶) has different length. Detailed sample dimension is shown in Table S1 (same as Table R1), including length, width, and thickness. **b** Measured thermal conductivity for Al films.

3) My numeric comment 3 was dismissed. Validate measurements with (opaque) films in the range of 30 W/m-K to 60 W/m-K. The authors state that validation was performed on stainless steel 304 (~15 W/m-K) and in Fig 2Ra it clearly has the steepest slope (because the conductance is ~2.1 W/K where the polymers in this study have conductance's around 0.4 W/K). Validation was also performed on Zylon and Dyneema but with conductance is around 1.3 W/K. I'd strongly suggest that when the requested validation measurements are performed, that the geometry is appropriately adjusted to maintain the same approximate overall conductance (0.4 W/K), as was conducted with the drawn films. (Side note: rather than adjusting the geometry, a more convincing study would be if the geometry was fixed (e.g., cut the samples to the same smallest size), as it would more clearly show how only the thermal conductivity was changing, not both geometry and conductivity. This would have been a better plan of experiments from the beginning.)

To further validate measurements on opaque films with thermal conductivity in the range of 30 W/m-K to 60 W/m-K, we chose Sn⁶ film as reference sample, because the literature thermal conductivity for Sn is ~67 W/m-K⁶ that is similar with that of 110 × drawn polyethylene films.

Fig. R2a (same as Fig. S1c) and Fig. R3a (same as Fig. S1a) show additional validation experiments using Sn samples⁶. We controlled the overall conductance ~ 0.4 W/K of all these Sn samples, which is similar with conductance of drawn polyethylene films (Fig. R2b and Fig. R2c, same as Fig. S1e and Fig. S1d). Fig. R2a (same as Fig. S1c) shows the measured electrical heating power versus temperature difference of a highly conductive Sn⁶ as a function of length. Fig. R4 (same as Fig. S1g) show the measured thermal conductivity ~60.47 W/m-K (Sn length 2.55 mm), ~64.4 W/m-K (Sn length 3.77 mm) , and ~65.3 W/m-K (Sn length 5.55 mm), which are all in good agreement with literature values⁶. We note that our measured thermal conductivity is relatively lower than the literature thermal conductivity, which is again due to the thermal interface resistance between the measured sample and the copper clamps during the measurements (Fig. R3a, same as Fig. S1a). We minimize thermal interface resistance by using thermal paste (Fig. S1), and we discussed thermal contact resistance effects in supplementary information 1.2.1 steady-state method minimization of thermal interface resistance. Because there is an effect of thermal contact resistance, we thus UNDERESTIMATED thermal conductivity of all the samples.

These additional measurements give us further confidence in our measurements of the polyethylene films and we hope the reviewer will be satisfied.

Figure R4. Measured thermal conductivity for Sn⁶ films with different geometries. Error bars indicate uncertain geometrical parameters. Detailed analysis of geometric uncertainties is discussed in SI 1.2 section: thermal conductivity measurements in the supplementary method.

5) My numeric comment 5 was dismissed. I must adamantly insist that these other measurements be performed: TDTR (with beam offset), 3-omega (with multi-heater lines width), transient grating, and/or laser flash (with film in-plane measurements). I am glad that the authors “are familiar with these techniques and have them in our [their] own lab...” I therefore expect that they should be able to perform the requested measurement quickly and with little cost burden. I am also intimately familiar with the merits and caveats of these techniques, which is why I explicitly suggested this subset of techniques. The authors have already performed TDTR experiments and report on the results for 50x in this manuscript (despite adhesion challenges), so that should be doable to perform on the higher conductivity samples (e.g., 100x). 3-omega can be insensitive to surface roughness and I find it hard to believe that the films are too rough (these are drawn polymer films after all).

Firstly, the accuracy of our measurements is validated by our measurements of carefully selected reference samples (Dyneema fiber⁵, Zylon fiber⁵, stainless steel film⁴, Sn⁶ film and Al⁶ film), our experimental results agree well with literature values^{4,5,6} (Fig. 2 and Fig. S1). Such accuracy is achieved via careful considerations and management of radiation and conduction heat losses, as well as radiation shunting between the hot and cold clamps.

Secondly, we thank Reviewer 1 comments on this point, and we agree with Reviewer 1 that some alternative methods are NOT a productive or necessary venture. Here is Reviewer 1’s detailed comments on this points:

“With regard to Reviewer 2’s comments: the 2nd reviewer suggested that techniques such as TDTR w beam offset or three omega should be used to confirm the homebuilt measurements. These techniques are very challenging for these samples, and would

probably not be better than a correct implementation of the current method. The 3-omega technique requires the ability to microfabricate a heater (and a rather thin heater if you want sensitivity to anisotropy; the assumption of constant heat flux is also questionable for a high k metal film on a low k substrate) and make electrical connections to it, rather hard when the metal film adhesion is low and the sample geometry is unconventional; this is quite challenging for polymeric/non-standard samples and may even alter the sample. Similarly the TDTR offset technique might be difficult on this sample because metal transducers themselves have a large in-plane conductance and the vertical thermal penetration depth of heat would necessarily be small for PE because of the low thermal conductivity in other directions. I have not examined the sensitivity analysis in this situation, but I do not think it would be favorable to beam offset TDTR. I definitely think that we should scrutinize the current thermal measurement method, but I do not think that some alternative method of experiments would be a productive or necessary venture.”

We respectfully disagree with Reviewer 2 comment “they should be able to perform the requested measurement quickly and with little cost burden”. We spent quite a few years to come up a method we are confident and know how much time each measurement takes. The corresponding author’s group had used pretty much all the methods the reviewer mentioned and know their limitations.

Transient grading may be challenging given opacity or adhesion, but I leave it as an option. In-plane laser flash is trivially easy and extremely well suited for this measurements and sample geometry given the claimed high thermal conductivity (and adherence to an ASTM standard). The commercial vendor Netzsch has a laboratory in proximity to the corresponding author’s lab and they would happily perform these measurements at no cost; I’m even willing to call Netzsch on the author’s behalf if necessary. Authors can use Netzsch thin-film holder, coat the sample with graphite paint, and even correct for transparent films effects. Measurements can be completed within hours at negligible cost.

We have transient grating platforms in our lab and through our collaborations. This technique cannot be readily applied due to poor sample absorption and roughness.

6) My numeric comment 6 was dismissed. Based on the data to be provided the amorphous thermal conductivity calculations may still need to be redone.

According to reviewer 2 comments, we did additional measurements on reference samples (Sn⁶ and Al⁶ films, Fig. S1). The measured thermal conductivity for all the reference samples and drawn polyethylene films are reliable and consistent^{4,5,6} (Fig.2, Fig S1 and S2). We hope these measurements have clarified the questions. At this time, we don’t see any value in re-doing any calculations.

I will not yield my position that this manuscript is not appropriate for publication until I see the requested data. These researchers are uniquely situated to perform these additional measurements since they are the only group with access to these drawn films. This characterization should be

performed for the betterment of the entire thermal science community, for which the corresponding author is an admired leader and trusted pillar. Very frankly, I would like to avoid another spider-silk or polyethylene-nanofiber high thermal conductivity measurement fiasco. Research resources are finite and this extra effort up-front could save others significant time not trying to reproduce erroneous results of high thermal conductivity.

Let us first be clear, the reviewer cited two entirely different papers^{9,10}: one is the polyethylene nanofiber⁹ from the corresponding author's group, and the other on the spider-silk¹⁰ which had NOTHING to do with the corresponding author's group. While the lead author of the spider-silk paper¹⁰, Dr. Xiaopeng Huang, is a co-author on the current paper as he participated in this research during his post-doc stay at the corresponding author's group, we relied on our own method which had been tested and validated before.⁷ The new data added are further testimony to the reliability of our method.

We stand by our previous work⁹ on polyethylene nanofiber also refer to the reviewer of a new paper published in *Nature Communications in 2018*¹¹ by the lead author of our previous polyethylene nanofiber, using a different fabrication and measurement methods¹¹.

Although we are not sure we can convince the reviewer with new data and this report, nor change the mind of the reviewer that our previous work was a "fiasco", we have done our best to address the reviewer's concerns. While we appreciate the reviewer's "frankness", we believe thoughtful, constructive, and respectful reviews are conducive to scientific progress.

Reviewer #3:

First I'd like to thank authors for clarifying their contribution, which is the "revealing of a surprisingly high thermal conductivity in the amorphous phase." I have a few further questions to verify this point. As the draw ratio increases, "amorphous" region develops orientation order. This is expected, as it is becoming more crystalline. Given this increasing order, is it still valid to term this region "amorphous"? On a related note, do the X-ray scattering measurements have sufficient signal-to-noise to be able to compare level of structural disorder in these regions to the structural disorder in bulk polymers commonly referred to as "amorphous"?

We appreciate that the referee captured this point. It is exactly this structurally modified "amorphous" region that leads to the interesting thermal properties. The term "amorphous" often refers to those non-crystalline and much disordered regions in a bulk material, where molecules have no preferred orientations. Upon mechanical stretching, molecules in an amorphous region may adopt certain orientations but remain in a non-crystalline state. There is no specific term in the polymer field to name this type of amorphous region, it is probably more appropriate to call it "oriented amorphous" in this work in order to distinguish from those totally randomly oriented amorphous regions.

In principle, X-ray scattering is able to distinguish oriented amorphous from random amorphous provided that the amorphous scattering which is originally a broad isotropic ring develops angular dependence. In reality, however, it is challenging because the amount of oriented amorphous may not be sufficient to scatter photons, and often the scattering angle of the amorphous peak is close to a strong low-order Bragg peak. However, we were still able to indirectly model this region during the SAXS analysis (as depicted in Fig. S10) because adopting orientation preference does not noticeably alter the average scattering length density in comparison with the crystalline region. With thermal modeling, we were still able to conclude a significant boost of thermal conductivity based on the statistical information about the sizes of the crystalline region and its neighboring regions (random amorphous and/or oriented amorphous) via the quantitative SAXS analysis.

The manuscript states that k_a is high compared to "typical 0.3 W/m-K", but wouldn't that 0.3 value be associated with films that have no orientation? If the orientational order here is increasing, why is region's increase in thermal conductivity surprising?

We think the surprising part lies in the fact that amorphous region itself can develop very high thermal conductivity while still being noncrystalline. Orientation effect has been traditionally attributed to the crystalline regions in semicrystalline polymers¹². While recently both experiment¹³ and simulation¹⁴ have suggested orientation can promote thermal transport in amorphous polymers as well, we observed a much larger enhancement. Considering this, such an enhancement in the amorphous region points to another venue to tune polymer's thermal transport and could stimulate further fundamental studies into such system.

The authors use a periodic 1D thermal model, including in their sensitivity analysis. Could the authors clarify how they know that the regions are always only in series and never in parallel? Figure 1d shows the regions as being perfect columns, or discs in cross-section. How sure are the authors that this is valid? This is important to verify given extremely large difference in thermal conductivity assumed for the two regions. If the crystalline regions were to penetrate the amorphous regions at all (for example, small pinholes that acted as thermal shunts), or if they were to tilt at an angle, so that a given cross-section of the fiber would have both crystalline and amorphous regions, the thermal model would appear to be invalid.

We thank the reviewer 3 for his/her comments. The WAXS analysis reveals that the (hk0) Bragg peaks are very narrow and are perpendicular to the drawing direction, indicating that the orientation of the crystallites is well aligned without much titling with respect to the drawing direction (Fig. 4 and Fig. S9c). In addition, SEM images show the interior nanofiber diameters is ~ 8 nm (Fig. S3c). Guinier-Porod analysis of the SAXS data along the direction perpendicular to the drawing indicates a mean diameter of nanofiber is ~ 11.1 nm (Fig. S11c), which is consistent with SEM images. The periodic intensity modulations along the drawing direction in the 2D SAXS pattern is a standard feature that reflects the lamellar structures along that drawn direction (Fig. 4c, Fig. 4e and Fig. S10). We analyze small-angle X-ray scattering data (SAXS) by statistically modeling the form and structure factors (supplementary information 1.3.3 SAXS analysis, structure factor and size distribution), and we quantify the alternating structures of “amorphous” and crystalline regions inside the fiber (schematically shown in Fig S10).

Since the individual nanofibers diameter was estimated to be ~ 10 nm (Fig. 1i-j, Fig. S3c and Fig. S11c) and the lamellar structures along that drawn direction (Fig. 4c, Fig. 4e and Fig. S10), it justifies the use of a one-dimensional model $k = [(1 - \eta)/k_c + \eta/k_a]^{-1}$ for the axial thermal conductivity (supplementary information 1.4 Thermal conductivity model). The amorphous region is an effective interpretation, as part of them might originate from the crystallites in the undrawn state during the drawing, but they are so stretched out that they can be no longer distinguished from other stretched amorphous chains.

Finally, I would also like to ask authors to consider revising their abstract and concluding paragraph (“In summary...”) to better reflect their stated contribution. As it currently reads, it appears they are saying that their contribution is the knowledge that the amorphous phase dictates overall thermal conductivity, which as they have agreed in their response is what one would normally expect.

We thank the reviewer comments. We revised the abstract and summary.

We attached marked manuscript to show where revisions were made.

Reference:

1. Kavesh, S. & Schultz, J. M. Meaning and measurement of crystallinity in polymers: a review. *Polym. Eng. Sci.* **9**, 452–460 (1969).
2. Kasai, N. & Masao, K. *X-ray diffraction by macromolecules*. (Springer, 2005).
3. Jiang, Z. GIXSGUI: a MATLAB toolbox for grazing-incidence X-ray scattering data visualization and reduction, and indexing of buried three-dimensional periodic nanostructured films. *J. Appl. Crystallogr.* **48**, 917–926 (2015).
4. Sweet, J. N., Roth, E. P. & Moss, M. Thermal conductivity of Inconel 718 and 304 stainless steel. *Int. J. Thermophys.* **8**, 593–606 (1987).
5. Wang, X., Ho, V., Segalman, R. A. & Cahill, D. G. Thermal conductivity of high-modulus polymer fibers. *Macromolecules* **46**, 4937–4943 (2013).
6. Arpaci, V. S., Kao, S.-H. & Selamet, A. *Introduction to Heat Transfer*. (Prentice Hall).
7. Kraemer, D. & Chen, G. A simple differential steady-state method to measure the thermal conductivity of solid bulk materials with high accuracy. *Rev. Sci. Instrum.* **85**, 025108 (2014).
8. Zhao, D., Qian, X., Gu, X., Jajja, S. A. & Yang, R. Measurement Techniques for Thermal Conductivity and Interfacial Thermal Conductance of Bulk and Thin Film Materials. *J. Electron. Packag.* **138**, 040802 (2016).
9. Shen, S., Henry, A., Tong, J., Zheng, R. & Chen, G. Polyethylene nanofibres with very high thermal conductivities. *Nat. Nanotechnol.* **5**, 251–255 (2010).
10. Huang, X., Liu, G. & Wang, X. New Secrets of Spider Silk: Exceptionally High Thermal Conductivity and Its Abnormal Change under Stretching. *Adv. Mater.* **24**, 1482–1486 (2012).
11. Shrestha, R. *et al.* Crystalline polymer nanofibers with ultra-high strength and thermal conductivity. *Nat. Commun.* **9**, Article number: 1664 (2018).

12. Choy, C. L. Thermal conductivity of polymers. *Polymer* **18**, 984–1004 (1977).
13. Singh, V. *et al.* High thermal conductivity of chain-oriented amorphous polythiophene. *Nat. Nanotechnol.* **9**, 384–390 (2014).
14. Liu, J. & Yang, R. Tuning the thermal conductivity of polymers with mechanical strains. *Phys. Rev. B* **81**, 174122 (2010).

Reviewers' comments:

Reviewer #1 (Remarks to the Author):

1. I can see that some personal issues have arisen between the author's and reviewer #2. While I criticized Reviewer #2's to-do list in the previous round of review, I would also confirm that the manuscript does have a "trust me" feel to it when reading, making it frustrating as a reviewer to discern whether an omission is just a mistake or is intentional. In virtually every aspect of this review that I have attempted replicating by hand using the supplied data and statements made in the manuscript, I have run into issues (see point #4 for example, but I've run into numerous issues also trying to use the data in Ref 18, and Reviewer #2 has run into numerous issues trying to understand the measurement method). As I said in a previous review, the plotted amorphous thermal conductivity at high draw ratio is probably inaccurate as well, even if the series model interpretation is correct. Sometimes these issues have ultimately turned out to be inconsequential to the final results, but every inconsistency chips away at confidence. I haven't found anything that is "definitely invalidating", but it can be difficult to discern when there is so much analysis going on under the hood.
2. For example, I've dug a little more deeply into the DFT analysis in Ref 18.

As part of this review, I have tried to replicate the authors calculations for their thermal model as well as a few competing models. I can see now that there is quite a bit of challenge to this (and missing data, see point #4). To do this, you would need to know the crystalline domain length (which the authors do, using scattering data, albeit only up to 50X draw) and the crystal fraction (which as I point out in #4 the authors are omitting some data for), and you need to know the thermal conductivity of crystalline phase as a function of boundary scattering length and chain length. On this last point, I got quite confused when trying to use the data from reference 18. I believe the plot being used by the authors is below (though it isn't adequately explained what data is being used or how from Ref. 18...Reference 18 does not contain any data related to boundary scattering. It contains an accumulation function with respect to mean-free-path, which is not quite the same thing since modes that exhibit boundary scattering should be treated with a different/finite MFP rather than "cutting off" the thermal conductivity.

Figure 6. (a) Variations of the axial thermal conductivity of infinite 1-D PE chain as a function of temperature, along with the absolute κ contributions from the four acoustic phonon modes and the κ values from the SMRT model; κ contributions from optical phonons are negligible and thus not shown. The convergence test at 300 K with increasing mesh size is also shown in the inset. (b) Variation of the axial thermal conductivity of 1-D PE chain at 300 K as a function of chain length l . Inset of (b), normalized cumulative axial thermal conductivity with the distribution of phonon mean free path A .

If I use the boundary lengths implied by Fig. S11b (the boundary lengths for the most important samples are of the scale $\sim 10\text{nm}$ - 20nm ; all of the others are smaller), which would put them in an illegibly small percentage of the cumulative thermal conductivity curve (inset) from Fig. 6 of Ref 18 (maybe 5% of the total if I'm being generous). To be blunt, I was just not able to replicate the numbers being used in the current manuscript.

3. I am very satisfied with the response given with regard to obtaining the crystalline fraction. In fact, the Figure provided to the reviewer (marked R1b) is vastly superior to the one currently provided in the SI material (the one in the SI is illegible), and proves the authors' point ably. I think the authors should replace the figure S9b with this (or the equivalent figure on a logarithmic plot).
4. I do see something weird about the crystallinity plot in the article file now, Fig 4d. This plot appears to have missing data points if I compare it to figure 3a&b. In order to obtain Fig. 3b, I believe the authors would have needed the crystallinity for all the points, but the crystallinity wasn't reported for 5 of the points (see my plot below). Was the crystallinity measured experimentally for these points? If so, please indicate the measurement values on the graph or explain how the values were obtained.

- In my first round of review, I argued that the combination of structural evolution data with thermal conductivity data is really the core discovery of the paper. Namely, the authors have discovered that crystalline alignment is not sufficient to explain the rise in total thermal conductivity, and that most of the thermal improvement occurs by the lengthening of the crystalline PE phase sandwiched by a shrinking amorphous phases (in a surprisingly superlattice-like arrangement according to Fig. 4e).

The authors argue in their next-to-last paragraphs of the paper that this is explained only by a shockingly high amorphous thermal conductivity, and the authors put a lot of emphasis on that interpretation in the abstract and throughout the review process. The authors don't present any competing models for this though, and I'm not sure that this is the only possible explanation. I agree with the 3rd reviewer that it seems that this could trivially be explained by a combined parallel-series model, where there are simultaneously lamella but also multiple high thermal conductivity paths. I believe that the structure I've sketched below would show up identically in your scattering data and

cannot be precluded. It would give a SAXS-obtained diameter consistent with the SEM diameter, which appears to me is the only data that has been cited as proof in the explanation to Reviewer #3.

As I say in point #7, I **do** think that there is excellent science being done in this manuscript, but I am currently not 100% confident of the final interpretation of the “high amorphous thermal conductivity” at this point, despite what admittedly appears to be an incredible amount of invested effort by the authors.

6. I do see that the control measurements and data presentation for the homebuilt measurement system have been improved significantly and have made a substantial attempt to address the points they had previously neglected with regard to Reviewer #2. I see no “red flags” regarding the stated accuracy of the total k measurements.
7. I do see a minor point within the rebuttal of Reviewer #2’s comments, which I do not agree with. I don’t understand how the authors have concluded that the contact resistance in the Al case is affecting the experiment; I would have thought this would manifest as a finite slope in Fig R3b, but I see no such slope. How did the authors come to that conclusion (I don’t think the Fig. numbers cited are correct as Fig. 3a says nothing about contact resistance)? But I suppose the real question is, does a 20% errorbar for aluminum (if the real thermal conductivity is actually 237 W/m-K) actually imply a failure of the homebuilt system to measure the thermal conductivity of the polymer films with the stated accuracy?

Reviewer #2 (Remarks to the Author):

First, I am very grateful to the authors for providing the requested data and for conducting the additional validation experiments. With this newly provided data, I have a greater confidence in the results now that I have seen the additional supporting data.

Thank you for replacing Figure 2b of the main text. I have independently verified the results from this data, which I was unable to do with the original Figure 2b.

Specifically, I used data-grabbing software to extract the raw data from this figure. From that data, I calculated the conductance and found it to be in good agreement with Table S1a. I then calculated the thermal conductivity from the geometric data provided in Table S1a and found it to also be in good agreement with the thermal conductivity data reported in Figure 3a. Specifically, my calculations were within 2% of the thermal conductivity reported in the manuscript for all draw ratios except 10 and 60, where I found 38% and -21% difference (which I will not contest further since it is within error bars). I will suggest that the authors may want to double check these two draw ratios (at their discretion). Originally, I was unable to reproduce the results with the data presented in the original Figure 2b, due to the geometrically scaled heating power confusion (which was previously discussed in the last round of reviews, and has since been remedied). Being able to reproduce the results has now greatly quieted my concerns of erroneous data; I am grateful to the authors for their patience and willingness to make this change.

Thank you for performing the additional control measurements on Sn and Al. This demonstrates that the homebuilt technique is suitable to measure higher thermal conductivity samples that was not performed in the group's previously cited publications. Again, I used data-grabbing software to extract this raw data from Figure 3c. I calculated the conductance and found it to be in good agreement with Table S1b (less than +/- 23% difference). I was further very satisfied to see that the parasitic heat loss (y-intercept of Figure 3c) in the control measurements were typically below <0.1 mW, just like the previous publications.

In comparison, the parasitic heat loss was (nominally) ~1 mW for the drawn polyethylene samples (which is 10%-20% of the full scale electrical heating power) in the main manuscript under review; this was my initial concern in the first round of reviews. Thank you for also providing the photographs of the Al and Sn samples; I have verified the geometries (i.e., length measurements) which was my original concern; I cannot explain the large parasitic heat loss through geometry errors. And thank you for using the Al and Sn samples as the opaque materials; that further quiets my original concern that the parasitic heat loss was due to radiation.

I do not have any remaining explanations or conjectures for why the parasitic heat loss is so much higher for the drawn polyethylene measurements than was reported for the control samples or in the group's previous cited work. I am satisfied that the large parasitic heat loss is now openly disclosed in the new Figure 2b of the main manuscript, rather than in the SI, even though I am unable to explain it. Thank you for openly disclosing this at the onset of the manuscript...maybe someone else can explain why the parasitic heat loss is so large for these measurements on drawn polyethylene.

I'll acquiesce on additional verification measurements, now that the control measurements show that the homebuilt technique is capable of measuring thermal conductivity/conductance in the appropriate range. I may contact the corresponding author in the future requesting one of these high conductivity samples and I'll perform the validation measurement myself at my own cost.

Reviewer #3 (Remarks to the Author):

The authors cite Choy's work from 1977 in saying that "Orientation effect has been traditionally

attributed to the crystalline regions in semicrystalline polymers." However, thermal conductivity anisotropy in spin-coated films has been studied for decades (for example, Goodson's work), with molecular orientation normally cited as reason for the large in-plane thermal conductivity. A quick web search found in-plane values approaching 2 W/mK for polyimide. Higher values may also be found in literature with a more thorough search.

When authors state that "our findings here will steer the research to a new track — engineering amorphous chains to achieve even higher thermal conductivity." I therefore do not see this as a new track. It's true that the "amorphous" values they derive are larger than 2 W/mK, however the underlying physics is the same. This work has importance in establishing a higher possible value for oriented chains at very high strain, but since there is no new understanding provided, I do not feel it is a good match to this journal.

Futhermore, values for the "amorphous" regions in this work are derived indirectly with limitations cited by other reviewers. By comparison, in-plane measurements of spin-coated films have very little potential for error.

Nanostructured Polymer Films with Metal-like Thermal Conductivity

We thank the reviewers for their constructive comments. Reviewers' inputs are very helpful for improving our manuscript. Our point-by-point responses are documented below. The reviewers' original comments are in black and our responses are in blue.

Reviewer #1

1. I can see that some personal issues have arisen between the author's and reviewer #2. While I criticized Reviewer #2's to-do list in the previous round of review, I would also confirm that the manuscript does have a "trust me" feel to it when reading, making it frustrating as a reviewer to discern whether an omission is just a mistake or is intentional. In virtually every aspect of this review that I have attempted replicating by hand using the supplied data and statements made in the manuscript, I have run into issues (see point #4 for example, but I've run into numerous issues also trying to use the data in Ref 18, and Reviewer #2 has run into numerous issues trying to understand the measurement method). As I said in a previous review, the plotted amorphous thermal conductivity at high draw ratio is probably inaccurate as well, even if the series model interpretation is correct. Sometimes these issues have ultimately turned out to be inconsequential to the final results, but every inconsistency chips away at confidence. I haven't found anything that is "definitely invalidating", but it can be difficult to discern when there is so much analysis going on under the hood.

Response 1:

We respect the reviewer's careful reading of the manuscript and comments, which has helped us to further improve the manuscript.

2. For example, I've dug a little more deeply into the DFT analysis in Ref 18. As part of this review, I have tried to replicate the authors calculations for their thermal model as well as a few competing models. I can see now that there is quite a bit of challenge to this (and missing data, see point #4). To do this, you would need to know the crystalline domain length (which the authors do, using scattering data, albeit only up to 50X draw) and the crystal fraction (which as I point out in #4 the authors are omitting some data for), and you need to know the thermal conductivity of crystalline phase as a function of boundary scattering length and chain length. On this last point, I got quite confused when trying to use the data from reference 18. I believe the plot being used by the authors is below (though it isn't adequately explained what data is

being used or how from Ref. 18...Reference 18 does not contain any data related to boundary scattering. It contains an accumulation function with respect to mean-free-path, which is not quite the same thing since modes that exhibit boundary scattering should be treated with a different/finite MFP rather than “cutting off” the thermal conductivity.

Figure 6. (a) Variations of the axial thermal conductivity of infinite 1-D PE chain as a function of temperature, along with the absolute κ contributions from the four acoustic phonon modes and the κ values from the SMRT model; κ contributions from optical phonons are negligible and thus not shown. The convergence test at 300 K with increasing mesh size is also shown in the inset. (b) Variation of the axial thermal conductivity of 1-D PE chain at 300 K as a function of chain length l . Inset of (b), normalized cumulative axial thermal conductivity with the distribution of phonon mean free path A .

If I use the boundary lengths implied by Fig. S11b (the boundary lengths for the most important samples are of the scale $\sim 10\text{nm}-20\text{nm}$; all of the others are smaller), which would put them in an illegibly small percentage of the cumulative thermal conductivity curve (inset) from Fig. 6 of Ref 18 (maybe 5% of the total if I'm being generous). To be blunt, I was just not able to replicate the numbers being used in the current manuscript.

We thank the reviewer for double checking the data, raise critical questions, and apologize for the confusion. We will address comments on crystalline domain length and crystal fraction in detail in Response 4 but only mention that for modeling, we relied only on SAXS data which give lengths of the crystalline and amorphous segments. Here, we will focus on explaining how we used Ref. 18¹ to extract the thermal conductivity of the crystalline domain, which is the basis for our claim of the high thermal conductivity of the amorphous phase.

First, the plot the reviewer cited from Ref 18¹ (Fig. 6b in the paper) is indeed the one we used. However, we want to point out that this figure shows the thermal conductivity of 1D PE chain as a function of the chain lengths rather than phonon mean free path, as is clearly indicated in the figure legend and caption.

Second, the reviewer is right that the crystalline domain thermal conductivity depends on both its length and lateral size. We choose data of 1D chain instead of PE crystals to estimate the thermal conductivity of the crystalline domain in our samples. The same manuscript actually solved Boltzmann transport equation assuming diffuse phonon scattering at all boundaries. If we used the values simulated (also kindly provided by Professor Baoling Huang¹), the amorphous region thermal would be even higher since the crystalline region thermal conductivity will be lower. In fact, the highest simulated crystalline region thermal conductivity corresponding to our crystallite size is ~40 W/m-K, lower than our measured total thermal conductivity. We believe assuming completely diffuse scattering at boundaries may not be accurate for two reasons: (1) phonon scattering at the lateral interface may not be diffuse since the interaction between the crystalline region and the surrounding is very weak (van der Waals interaction), and (2) there is a transition from the crystalline region to the amorphous region along a single fiber, and some molecules may be continuous from the fiber to the amorphous region. Due to these reasons, we used simulated thermal conductivity of 1D chains with finite length, which have higher thermal conductivity (compare Fig. 6b and Fig. 4a¹). This choice will only underestimate the thermal conductivity in the amorphous region.

Regarding the figure reading issue, we have requested the original data for Fig. 6b from Professor Huang's group¹. They also kindly provide new simulation data on two smaller chain lengths (4 and 5nm) to cover the entire range of crystallite sizes we observed. Red dots in Fig. R1 represent the original data from Professor Huang's group¹ (up to 50nm). To obtain the relationship between crystalline thermal conductivity and the draw ratio in our study, we first find the crystallite size at given draw ratio based on Fig. S15b, and then interpolate the data in Fig. R1 to get the corresponding crystalline thermal conductivity, taking the chain length as the crystallite size. We note that the updated plot of crystalline thermal conductivity with respect to the draw ratio (Fig. S15c) is very similar to what we had before.

Figure R1. Thermal conductivity (along the chain direction) as a function of the 1D polyethylene chain length from Wang et al.'s reference¹ and our fitting results. Original data (dot) is from reference¹, and the curve is from our fitting results.

Revisions made: We have added Fig. R1 in SI (Fig. S15a, same as Fig. R1) and tried to make it more clear in our writing how the crystalline phase thermal conductivity is modeled. The quadratic function used to fit the simulation data is provided in SI section 1.4.1. The crystalline thermal conductivities and the amorphous thermal conductivities with respect to the draw ratio are also updated accordingly (Fig. 3b and Fig. S15c). Since the amorphous region thermal conductivity represent a lower limit based on our model, we have removed the statement that the amorphous region is where the major resistance lies.

3. I am very satisfied with the response given with regard to obtaining the crystalline fraction. In fact, the Figure provided to the reviewer (marked R1b) is vastly superior to the one currently provided in the SI material (the one in the SI is illegible), and proves the authors' point ably. I think the authors should replace the figure S9b with this (or the equivalent figure on a logarithmic plot).

Response 3:

Thank you, we have revised the supporting information and marked both R1b and R1a results in Fig. S9.

4. I do see something weird about the crystallinity plot in the article file now, Fig 4d. This plot appears to have missing data points if I compare it to figure 3a&b. In order to obtain Fig. 3b, I believe the authors would have needed the crystallinity for all the points, but the crystallinity wasn't reported for 5 of the points (see my plot below). Was the crystallinity measured experimentally for these points? If so, please indicate the measurement values on the graph or explain how the values were obtained.

Response 4:

We respect that the reviewer checked our data. We did not perform WAXS measurements on every draw ratio. However, please note that the crystallinity results (Fig. 4d WAXS) is used only for understanding the thermal conductivity trend, it is NOT for obtaining Fig. 3b results. As we noted in the Figure 3b caption “Extracted amorphous thermal conductivity values based on fitted structural parameters from **SAXS analysis**”.

Here is the reason why SAXS are used for thermal model in Fig. 3b. This is because SAXS results can confirm critically lamellar superlattice structure (Fig. S10) and provide the amorphous and crystalline length (Fig. S11). The WAXS cannot provide these information. Having lamellar superlattice structure and crystalline/amorphous length information are central to deriving results in Fig. 3b. However, we would like to point out that the WAXS data on crystallinity is in general agreement with SAXS data based on the ratio of the length of the amorphous region and crystalline region. To show this, we have converted the WASX crystallinity data to “amorphous/total length”

ratio and plotted them in Fig.4f. This addition helps to increase our confidence since, as the reviewer pointed out, our SAXS data is only to draw ratio of 50. Beyond which, the superlattice period length is too long to be resolved by the SAXS data.

With regard to how the length of the amorphous and crystalline regions are obtained, we have described them in detail in both manuscript and SI 1.3-1.4 (Fig. 4c-f, Fig. S10-S11, manuscript and SI). Here we briefly re-summarize the procedure we follow to obtain the crystalline domain length and crystal fraction in comments #2:

The periodic unit length is obtained by SAXS analysis (SI 1.3.3). The fraction of the amorphous region is determined by partitioning the low density region from the electron density profile based on SAXS measurement (Fig. 4f, SI 1.4.1). One minus the amorphous fraction then gives crystal fraction. The crystalline domain length is calculated by multiplying the crystal fraction with the periodic unit length. We further interpolate these structural variables with draw ratios using smooth functions (SI 1.4.1). The thermal conductivities in the amorphous region (Fig. 3b) are finally calculated using the interpolated structural variables.

Revisions made: we have converted WAXS data in Fig. 4d into the amorphous/length ratio and add these data in Fig. 4f. This addition shows that the SAXS and WAXS data are consistent with each other and they are also within the bounds we assumed. This addition lends support for the modeling up to 110X draw ratio, even though SAXS data can only resolve superlattice length up to 50 draw ratio.

5. In my first round of review, I argued that the combination of structural evolution data with thermal conductivity data is really the core discovery of the paper. Namely, the authors have discovered that crystalline alignment is not sufficient to explain the rise in total thermal conductivity, and that most of the thermal improvement occurs by the lengthening of the crystalline PE phase sandwiched by a shrinking amorphous phases (in a surprisingly superlattice-like arrangement according to Fig. 4e).

The authors argue in their next-to-last paragraphs of the paper that this is explained only by a shockingly high amorphous thermal conductivity, and the authors put a lot of emphasis on that interpretation in the abstract and throughout the review process. The authors don't present any competing models for this though, and I'm not sure that this is the only possible explanation. I agree with the 3rd reviewer that it seems that this could trivially be explained by a combined parallel-series model, where there are simultaneously lamella but also multiple high thermal conductivity paths. I believe that the structure I've sketched below would show up identically in your scattering data and cannot be precluded. It would give a SAXS-obtained diameter consistent with the SEM diameter, which is appears to me is the only data that has been cited as proof in the explanation to Reviewer #3.

As I say in point #7, I **do think that there is excellent science being done in this manuscript**, but I am currently not 100% confident of the final interpretation of the “high amorphous thermal conductivity” at this point, despite what admittedly appears to be an incredible amount of invested effort by the authors.

Response 5:

Both this reviewer and Reviewer 3 raised a good point. Below we show that heat conduction along the curved path is **much smaller** than what would be required by our experimentally determined thermal conductivity. Therefore, this curved path is not the dominant heat conduction channel.

Figure R2. Schematic for heat flow along the curved path (start at fiber A, flow into surrounding fibers, and then back to fiber A). The path is drawn for one fiber in the surrounding but we note that our following analysis takes into account the case that multiple fibers can exist.

In Fig. R2, we show the schematic of the heat flow along the curved path. We will focus on one single fiber (fiber A), and estimate the thermal resistance of heat flow from fiber A to the surrounding fibers and then back to fiber A. We consider path starting at red dashed line and ends at blue dashed line, with a total length corresponding to one repeated unit.

For estimation, we take geometry data for the 50× drawn sample (crystalline thermal conductivity $k_c \sim 70\text{W/m-K}$, repeated unit length $L_{\text{tot}} \sim 22\text{nm}$, crystalline domain length $L_c \sim 20\text{nm}$) and take the nanofiber diameter D to be 10nm. Because the nanofibers assemble into bundles mostly via van der Waals interactions (our SAXS analysis (Fig S11c) showed similar diameter to the SEM (Fig S3c)), large interfacial thermal resistance will occur. We use an interfacial thermal conductance h of $3 \times 10^7\text{W/m}^2\text{K}$ based on literature data² for clean interface with van der Waals bonding. Because the interface in our case is not atomically flat, the actual thermal conductance can be even lower.

We estimate the thermal resistance from fiber A to surrounding fibers using a heat transfer fin model assuming that the fiber exchanges heat with surroundings with a heat transfer coefficient h . This model treats all neighboring fibers as a uniform environment, representing the worst case of multiple fibers closely packed with an infinite thermal conductivity. The thermal resistance between the starting point in fiber A and the surrounding is approximately

$$R_{fin,eff} \approx \frac{1}{\sqrt{k_c A h P} \cdot \tanh(m L_c / 2)}$$

where $A = \pi D^2 / 4$, $P = \pi D$, and $m = \sqrt{\frac{hP}{k_c A}}$ is the fin parameter. This leads to $R_{fin,eff} \sim 1.04 \times 10^8$ K/W. As the curved path involves passing interfaces twice, the total thermal resistance will be $2R_{fin,eff} \sim 2.1 \times 10^8$ K/W.

If the curved path were the dominant heat conduction channel, the corresponding thermal conductivity based on the fiber cross-sectional area and repeated unit length of the nanofiber should match our experimental measurement. However, the corresponding thermal conductivity for the heat flow along the curved path is

$$k_{tot} = \frac{L_{tot}}{A} \frac{1}{2R_{fin,eff}} = 1.3 \text{ W/m-K}$$

which is much smaller than the measured thermal conductivity (~ 30 W/m-K at $50\times$).

We conclude that the curved path has negligible contribution to the total heat conduction.

Revisions made: We have added the above response to the manuscript.

6. I do see that the control measurements and data presentation for the homebuilt measurement system have been improved significantly and have made a substantial attempt to address the points they had previously neglected with regard to Reviewer #2. I see **no "red flags"** regarding the stated accuracy of the total k measurements.

Response 6:

Thank you.

7. I do see a minor point within the rebuttal of Reviewer #2's comments, which I do not agree with. I don't understand how the authors have concluded that the contact resistance in the Al case is affecting the experiment; I would have thought this would manifest as a finite slope in Fig R3b, but I see no such slope. How did the authors come to that conclusion (I don't think the Fig. numbers cited are correct as Fig. 3a says nothing about contact resistance)? But I suppose the real question is, does a 20% errorbar for aluminum (if the real thermal conductivity is actually 237 W/m-K) actually imply a failure of the homebuilt system to measure the thermal conductivity of the polymer films with the stated accuracy?

Response 7:

The reason that the contact resistance does not lead to an apparent slope in the thermal conductivity is that the contact resistance is only a fraction of the thermal resistance of the heat conduction. Below we clarify this point using the model we presented in SI (section 1.2.1, Eq. 5-6).

The heat flux through the sample is $q = \frac{k}{l}(T_{h,s} - T_{c,s})$, where k is aluminum's thermal conductivity (237W/m-K)³, l is the sample length, and $T_{h,s}$ ($T_{c,s}$) is the sample temperature at the end of the clamp at hot (cold) side. The sample temperatures at the end of clamps are different from the hot/cold block temperatures (measured with thermocouples attached onto the blocks) due to the heat spreading and contact resistance. The difference can be estimated using a fin model, by extending Eq.(6) in SI to specifically include the thermal interfacial resistance between the film and the clamps caused by silver epoxy paste:

$$T_h - T_{h,s} = T_{c,s} - T_c = \frac{ql}{\sqrt{2kK_{CP}}}$$

where the effective cross-plane thermal conductivity K_{CP} lumps the heat spreading from the sample end to the copper clamp (governed by the cross-plane thermal conductivity k_{CP}) and the contact resistance R_C (t is the sample thickness):

$$K_{CP} = \frac{1}{2R_c/t + 1/k_{CP}}$$

The sample thermal conductivity is eventually derived based on T_h and T_c :

$$k_{measurement} = \frac{ql}{T_h - T_c} = \frac{1}{1 + \frac{d}{l} \sqrt{\frac{2k}{K_{CP}}}}$$

We found that with $K_{CP} = 0.3$ W/mK the value of $k_{measurement}$ matches our experiment. For metal films the dominant resistance comes from the interfacial contact instead of the heat spreading, namely $K_{CP} \approx \frac{1}{2R_c/t}$. Therefore, this K_{CP} value translates to a contact resistance of $R_C = 4.2 \times 10^{-5}$ m²K/W, which is equivalent to a 42 μ m layer thick silver epoxy paste if the uncured silver epoxy has a thermal conductivity of 1 W/mK and is reasonable. If we assume this contact resistance does not change, the measured thermal conductivity should vary with the sample length as shown below.

Figure R3. Blue curve represents expected thermal conductivity of aluminum from measurement with respect to the sample length, considering the contact resistance. The cross-sectional area takes the sample geometry (width is 0.51mm and thickness is 25 μ m). Red star represents measured thermal conductivity for Al films with different geometry (Table S1b). Error bars indicate uncertain geometrical parameters.

For the two aluminum samples with length 4.3mm and 5.79mm, the corresponding thermal conductivities are very close (192W/m-K and 202W/m-K) and within the error bar. Therefore, we argue that the lack of an apparent slope does not contradict the existence of the contact resistance.

Revisions made: We revise Fig. S1h by adding expected thermal conductivity of aluminum from measurement with respect to the sample length (considering the contact resistance). Fig. S1h is the same as Fig. R3. We also revise Fig. S1h caption with more detailed explanation.

Reviewer #2

1. First, I am very grateful to the authors for providing the requested data and for conducting the additional validation experiments. With this newly provided data, I have a greater confidence in the results now that I have seen the additional supporting data.

Thank you for replacing Figure 2b of the main text. I have independently verified the results from this data, which I was unable to do with the original Figure 2b. Specifically, I used data-grabbing software to extract the raw data from this figure. From that data, I calculated the conductance and found it to be in good agreement with Table S1a. I then calculated the thermal conductivity from the geometric data provided in Table S1a and found it to also be in good agreement with the thermal conductivity data reported in Figure 3a. Specifically, my calculations were within 2% of the thermal conductivity reported in the manuscript for all draw ratios except 10 and 60, where I found 38% and -21% difference (which I will not contest further since it is within error bars). I will suggest that the authors may want to double check these two draw ratios (at their discretion). Originally, I was unable to reproduce the results with the data presented in the original Figure 2b, due to the geometrically scaled heating power confusion (which was previously discussed in the last round of reviews, and has since been remedied). Being able to reproduce the results has now greatly quieted my concerns of erroneous data; I am grateful to the authors for their patience and willingness to make this change.

Response 1:

We are glad that the reviewer is satisfied. Since the reviewer did not give details with regards to 10 and 60 draw ratios, we can only reaffirm that our data is accurate and they are 12 and 27 W/m-K for 10× and 60× samples, respectively. We do not know how he/she got the difference for our 10 and 60 samples. We can only speculate, as our response to comment No.2 shows, that the reviewer's data grabbing may be not accurate. In addition to the above speculation of data grabbing, we want to emphasize the following:

- According to your request in the 2nd review round, we measured calibration tests for Sn and Al samples³. Fig. S1f shows the calibration for Sn and Al samples, it is not for 10×, 60× polymer samples.
 - As already clearly mentioned in both manuscript and SI, “we did calibration measurements, which directly measured the radiative thermal shunting after each sample measurement...” Note each measurement has calibration uncertainty, because manually mounting can cause slightly different view factors between the heater and cooler, and different radiative thermal shunting (Fig. S1a). For each sample, we used its own calibration data.
2. Thank you for performing the additional control measurements on Sn and Al. This demonstrates that the homebuilt technique is suitable to measure higher thermal conductivity samples that was not performed in the group's previously cited publications. Again, I used data-grabbing software to extract this raw data from Figure 3c. I calculated

the conductance and found it to be in good agreement with Table S1b (less than +/- 23% difference). I was further very satisfied to see that the parasitic heat loss (**y-intercept** of Figure 3c) in the control measurements were typically below <0.1 mW, just like the previous publications.

In comparison, the parasitic heat loss was (nominally) ~1 mW for the drawn polyethylene samples (which is 10%-20% of the full scale electrical heating power) in the main manuscript under review; this was my initial concern in the first round of reviews. Thank you for also providing the photographs of the Al and Sn samples; I have verified the geometries (i.e., length measurements) which was my original concern; I cannot explain the large parasitic heat loss through geometry errors. And thank you for using the Al and Sn samples as the opaque materials; that further quiets my original concern that the parasitic heat loss was due to radiation.

Response 2:

We respect that the reviewer has checked our data. However, our parasitic heat loss data for control measurements in Fig S1c (following Figure) have been misread. We note that the y-intercepts are NOT at the origin of Fig S1c (see pointers in the Figure). All the parasitic heat losses are ~1mW.

3. I do not have any remaining explanations or conjectures for why the parasitic heat loss is so much higher for the drawn polyethylene measurements than was reported for the control samples or in the group's previous cited work. I am satisfied that the large parasitic heat loss is now openly disclosed in the new Figure 2b of the main manuscript, rather than in the SI, even though I am unable to explain it. Thank you for openly disclosing this at the onset of the manuscript...maybe someone else can explain why the parasitic heat loss is so large for these measurements on drawn polyethylene.

I'll acquiesce on additional verification measurements, now that the control measurements show that the homebuilt technique is capable of measuring thermal conductivity/conductance in the appropriate range. I may contact the corresponding author in the future requesting one of these high conductivity samples and I'll perform the validation measurement myself at my own cost.

Response 3: We are glad that the reviewer seems to be satisfied with our response.

Reviewer #3

1. The authors cite Choy's work from 1977 in saying that "Orientation effect has been traditionally attributed to the crystalline regions in semicrystalline polymers." However, thermal conductivity anisotropy in spin-coated films has been studied for decades (for example, Goodson's work), with molecular orientation normally cited as reason for the large in-plane thermal conductivity. A quick web search found in-plane values approaching 2 W/mK for polyimide. Higher values may also be found in literature with a more thorough search.

Response 1:

We note pioneering work that shows ~ 2 W/m-K for polyimide film⁴ and ~ 4.4 W/mK⁵ for polythiophene nanofibers (diameter ~ 40 nm) and have cited these publications in our manuscript. To our knowledge, thermal conductivity over 10 W/m-K for amorphous chains in bulk polymers has not yet been demonstrated experimentally. We would be grateful if the reviewer could point to us where we can find such reports, if existing.

2. When authors state that "our findings here will steer the research to a new track — engineering amorphous chains to achieve even higher thermal conductivity." I therefore do not see this as a new track. It's true that the "amorphous" values they derive are larger than 2 W/mK, however the underlying physics is the same. This work has importance in establishing a higher possible value for oriented chains at very high strain, but since there is no new understanding provided, I do not feel it is a good match to this journal. Furthermore, values for the "amorphous" regions in this work are derived indirectly with limitations cited by other reviewers. By comparison, in-plane measurements of spin-coated films have very little potential for error.

Response 2:

We respectfully do not agree with the reviewer. We believe that the careful characterization, combined with the modeling and experiment, provides new understanding that the amorphous chains have high thermal conductivity. We acknowledge that the thermal conductivity values of the amorphous regions are derived, but they are based on solid structural analysis and best existing simulations and modeling. The derived high thermal conductivity of the amorphous region is, according to the analysis we presented, a conservative estimate. The reviewer's argument that the physics of high thermal conductivity is the same might be right, but the limits of thermal conductivity values of polymers are unresolved issues. Our work brings the amorphous polymer thermal conductivity to a new height.

Reference

1. Wang, X., Kaviany, M. & Huang, B. Further improvement of lattice thermal conductivity from bulk crystalline to 1-D-chain polyethylene: a high-yet-finite thermal conductivity using first-principles calculation. *ArXiv170102428 Cond-Mat* (2017).
2. Losego, M. D., Grady, M. E., Sottos, N. R., Cahill, D. G. & Braun, P. V. Effects of chemical bonding on heat transport across interfaces. *Nat. Mater.* **11**, 502–506 (2012).
3. Arpaci, V. S., Kao, S.-H. & Selamet, A. *Introduction to Heat Transfer*. (Prentice Hall).
4. Kurabayashi, K. & Goodson, K. E. Impact of molecular orientation on thermal conduction in spin-coated polyimide films. *J. Appl. Phys.* **86**, 1925–1931 (1999).
5. Singh, V. *et al.* High thermal conductivity of chain-oriented amorphous polythiophene. *Nat. Nanotechnol.* **9**, 384–390 (2014).

REVIEWERS' COMMENTS:

Reviewer #1 (Remarks to the Author):

With the exception of rebuttal of point #5, I think the authors have addressed my comments ably. I thank the authors for their extensive efforts addressing my and the other authors questions. On point #5, I think the fin approximation is questionable (the associated Biot number is at least of order 1, and I think the authors assumptions about the value of interface conductance are potentially too low), but I can see that this is a conservative approximation in the sense that a conduction-based resistance associated with the low thermal conductivity direction resistance would not fundamentally change their conclusion (potentially even making their argument stronger). As such I'm satisfied with the manuscript, and would recommend it for publication.

Reviewer #2 (Remarks to the Author):

I am grateful for the authors confirming the measured values at the 10x and 60x sample. I manually double checked the data-grab and I think it is accurate. Qualitatively, the 10x draw ratio (downward purple triangles in Fig 2b) shows a lower parasitic heat loss than any of the others (zooming in it is slightly below 0 mW), and 60x draw ratio (maroon stars) have a larger measured error bar for the first data point (at a temperature difference just above 2 K). Again, I don't feel the need to press this issue as the value that I calculated is within the error bars reported. Those two specific draw ratios just stood out from the others when I tried to replicate the analysis and I wanted to simply report that difference.

The authors are also correct regarding the origin on the controls. I looked at the origin obtained by the data-grabbing software and it is indeed at the wrong location. Screen shot attached for the control sample data grab. Thank you for identifying this. In hindsight, I should have just asked the Editor for you to provide the raw data in table form for verification rather than using data-grabbing software.

Measured electrical heating power, P_{el} (mW)

Revision Report for NCOMMS-17-22345C

Nanostructured Polymer Films with Metal-like Thermal Conductivity

Yanfei Xu, Daniel Kraemer, Bai Song, Zhang Jiang, Jiawei Zhou, James Loomis, Jianjian Wang,

Mingda Li, Hadi Ghasemi, Xiaopeng Huang, Xiaobo Li and Gang Chen*

e-mail: gchen2@mit.edu

We thank the reviewers for their valuable comments. Reviewers' inputs are helpful for improving our manuscript. Our point-by-point responses are documented below. The reviewers' original comments are in black and our responses are in blue.

Point-to-Point Response to Reviewer 1's Comments

Reviewer #1 (Remarks to the Author):

With the exception of rebuttal of point #5, I think the authors have addressed my comments ably. I thank the authors for their extensive efforts addressing my and the other authors questions. On point #5, I think the fin approximation is questionable (the associated Biot number is at least of order 1, and I think the authors assumptions about the value of interface conductance are potentially too low), but I can see that this is a conservative approximation in the sense that a conduction-based resistance associated with the low thermal conductivity direction resistance would not fundamentally change their conclusion (potentially even making their argument stronger). As such I'm satisfied with the manuscript, and would recommend it for publication.

Reply:

We thank the reviewer for his/her recommendation, and agree with him/her that our estimation has based on conservative approximation. Further, we want to clarify that, using the parameters we provided ($h = 3e7 \text{ W/m}^2\text{K}$, $k_c = 70 \text{ W/mK}$, and $D = 10 \text{ nm}$), the Biot number ($Bi = h*D/k_c = 0.0043$) is much less than 1 and the fin approximation is thus justified. We do acknowledge that the estimation of interface conductance may have uncertainty, but a value at the larger end (e.g. $h = 1e8 \text{ W/m}^2\text{K}$) only leads to a modest increase in the estimated total thermal conductivity ($k_{tot} \sim 4.3 \text{ W/mK}$), which is still far from being able to explain our data (measured value of $\sim 30 \text{ W/mK}$ at $50\times$).

Revision made: we added the above to the discussion on the fin model (Supplementary Note 5. thermal conductivity model).

Point-to-Point Response to Reviewer 2's Comments

Reviewer #2 (Remarks to the Author):

I am grateful for the authors confirming the measured values at the 10x and 60x sample. I manually doubled checked the data-grab and I think it is accurate. Qualitatively, the 10x draw ratio (downward purple triangles in Fig 2b) shows a lower parasitic heat loss than any of the others (zooming in it is slightly below 0 mW), and 60x draw ratio (maroon stars) have a larger measured error bar for the first data point (at a temperature difference just above 2 K). Again, I don't feel the need to press this issue as the value that I calculated is within the error bars reported. Those two specific draw ratios just stood out from the others when I tried to replicate the analysis and I wanted to simply report that difference.

The authors are also correct regarding the origin on the controls. I looked at the origin obtained by the data-grabbing software and it is indeed at the wrong location. Screen shot attached for the control sample data grab. Thank you for identifying this. In hindsight, I should have just asked the Editor for you to provide the raw data in table form for verification rather than using data-grabbing software.

Reply:

We are glad that the reviewer is satisfied. As the reviewer pointed out, the data are all within error bars. Hence, we take no further action.